# Intragenic recruitment of NF-κB drives splicing modifications upon activation by the oncogene Tax of HTLV-1

Lamya Ben Ameur [1], Paul Marie [1,8], Morgan Thenoz[1,2,8], Guillaume Giraud[1,8], Emmanuel Combe [1], Jean-Baptiste Claude[1], Sebastien Lemaire[1], Nicolas Fontrodona[1], Hélène Polveche[3], Marine Bastien[1,4], Antoine Gessain[5], Eric Wattel [6,7], Cyril F. Bourgeois [1], Didier Auboeuf [1✉] & Franck Mortreux [1✉]

Chronic NF-κB activation in inflammation and cancer has long been linked to persistent activation of NF-κB–responsive gene promoters. However, NF-κB factors also massively bind to gene bodies. Here, we demonstrate that recruitment of the NF-κB factor RELA to intragenic regions regulates alternative splicing upon NF-κB activation by the viral oncogene Tax of HTLV-1. Integrative analyses of RNA splicing and chromatin occupancy, combined with chromatin tethering assays, demonstrate that DNA-bound RELA interacts with and recruits the splicing regulator DDX17, in an NF-κB activation-dependent manner. This leads to alternative splicing of target exons due to the RNA helicase activity of DDX17. Similar results were obtained upon Tax-independent NF-κB activation, indicating that Tax likely exacerbates a physiological process where RELA provides splice target specificity. Collectively, our results demonstrate a physical and direct involvement of NF-κB in alternative splicing regulation, which significantly revisits our knowledge of HTLV-1 pathogenesis and other NF-κB-related diseases.

[1] Laboratory of Biology and Modelling of the Cell, Univ Lyon, ENS de Lyon, Univ Claude Bernard, CNRS UMR 5239, INSERM U1210, 46 Allée d'Italie Site Jacques Monod, 69007 Lyon, France. [2] Department of Pediatrics and Medical Genetics, Faculty of Medicine and Health Sciences, 9000 Gent, Belgium. [3] CECS, I-Stem, Corbeil-Essonnes 91100, France. [4] School of Pharmacy and Biomolecular Sciences, Liverpool John Moores University, Liverpool, UK. [5] Unité d'Epidémiologie et Physiopathologie des Virus Oncogénes, Institut Pasteur, Paris, France. [6] Université Lyon 1, CNRS UMR5239, Oncovirologie et Biothérapies, Faculté de Médecine Lyon Sud, ENS - HCL, Pierre Bénite, France. [7] Université Lyon 1, Service d'Hématologie, Pavillon Marcel Bérard, Centre Hospitalier Lyon-Sud, Pierre Bénite, France. [8] These authors contributed equally: Paul Marie, Morgan Thenoz, Guillaume Giraud. ✉email: didier.auboeuf@inserm.fr; franck.mortreux@ens-lyon.fr

The human T-cell leukemia virus (HTLV-1) is the etiologic agent of numerous diseases, including adult T-cell leukemia/lymphoma (ATLL)[1], an aggressive CD4$^+$ T-cell malignancy, and various inflammatory diseases, such as HTLV-1-associated myelopathy/tropical spastic paraparesis (HAM/TSP)[2]. Changes at the gene expression level participate in the persistent clonal expansion of HTLV-infected CD4$^+$ and CD8$^+$ T-cells, leading ultimately to HTLV-1-associated diseases[3]. We recently reported that alternative splicing events help to discriminate between ATLL cells, untransformed infected cells, and their uninfected counterparts derived from patients[4]. Alternative splicing of pre-messenger RNAs is a co-transcriptional processing step that controls diversity in both the transcriptome and the proteome and that governs cell fate. Its regulation relies on a complex and still incompletely understood interplay between splicing factors, chromatin regulators, and transcription factors[5,6]. For instance, we still do not fully understand the molecular mechanisms of HTLV-1–induced splicing modifications, or whether these effects rely on interplay between transcription and splicing.

Tax is an HTLV-1-encoded protein that regulates viral and cellular gene transcription. Tax also alters host signaling pathways that sustain cell proliferation, leading ultimately to cell immortalization[7]. The nuclear factor-κB (NF-κB) signaling pathway is the most critical target of Tax for cell transformation[8]. The NF-κB transcription factors (RELA, p50, c-Rel, RelB, and p52) govern immune functions, cell differentiation, and proliferation[9]. NF-κB activation involves the degradation of IκB, which sequesters NF-κB factors in the cytoplasmic compartment; this leads to NF-κB nuclear translocation and binding of NF-κB dimers (of which RELA:p50 is the most abundant) to their target promoters[10,11]. Tax induces IKK phosphorylation and IκB degradation, leading to persistent nuclear translocation of NF-κB[12,13]. In addition, Tax interacts with nuclear NF-κB factors and enhances their effects on transcription[14,15].

Interestingly, genome-wide analyses of NF-κB distribution have unveiled that the vast majority of RELA peaks is outside promoter regions and can be localized in introns and exons[16–19]. Some of those promoter-distant RELA-binding sites correspond to cis-regulatory transcriptional elements[20,21] but globally, there is a weak correlation between the binding of RELA to genes and regulation of their steady-state expression[17,18]. These data suggest that NF-κB could have other functions than its initially described function as a transcription factor.

Here, we show that NF-κB activation accounts for alternative splicing modifications generated upon Tax expression. These effects rely on a tight physical and functional interplay between RELA and the DDX17 splicing factor, which dynamically occurs upon NF-κB activation. Our results reveal that RELA DNA binding in the vicinity of genomic exons regulates alternative splicing through DDX17 recruitment, which then modulates exon inclusion via its RNA helicase activity.

## Results

### Tax regulates splicing regardless of its transcriptional effects.
We performed RNA-seq analyses on 293T-LTR-GFP cells transiently transfected with a Tax expression vector. We then identified Tax-induced changes in gene expression level and in alternative splicing and annotated them as previously described[22,23] (Supplementary Data 1 and 2). Notably, the ectopic expression of Tax affected the splicing and gene expression levels of 939 and 523 genes, respectively (Fig. 1a). A total of 1108 alternative splicing events was predicted, including 710 exon skipping events (Fig. 1b). A small percentage of genes (3.5%, 33/939) was altered at both their expression and splicing levels,

indicating that Tax largely affects alternative splicing independently of its transcriptional activity. We validate a subset of splicing events by RT-PCR (Fig. 1c). To test whether Tax-induced splicing modifications occurred in vivo, we took advantage of a publicly available RNA-seq dataset (EGAS00001001296[24]) from three healthy control donors, three asymptomatic carriers (AC), and 55 ATLL patients (Supplementary Data 1). Although lowly expressed, Tax mRNA was detected in the vast majority of infected samples (Supplementary Fig. 1). It is important to note that ATL can sporadically express Tax in bursts[25,26]; thereby explaining, at least in part, their Tax-like features even though Tax is poorly detected in ATL samples[24]. Based on this, we examined the percent spliced-in (PSI) values of exons expressed in samples from donors, carriers, and patients with ATLL (Supplementary Data 1). Overall, 542 (48%) Tax-induced splicing modifications were detected at least once in 55 ATL samples (Supplementary Data 1). This low frequency was anticipated due to the high fluctuations between individuals of Tax expression levels, proviral loads, and infected cell clonality, as well as somatic alterations (e.g., gene mutations, deletions, and duplications), which accumulate in ATLL cells[24]. Some Tax-related splicing events were recurrently detected in asymptomatic and ATLL samples (Fig. 1d). In addition, using exon-specific RT-PCR assays, we confirmed that Tax promotes splicing events that have been previously described in HTLV-1-infected individuals, including in the AASS, CASK, RFX2, and CD44 genes[4,27] (Fig. 1c). Altogether, these results uncovered a large number of splicing modifications that occurred upon Tax expression, which to a degree coincide with alternative splicing events observed in HTLV-1 patients.

### DDX17 interacts with RELA upon Tax-induced NF-κB activation.
As Tax is a well-known trans-acting transcription regulator, we first analyzed whether Tax could affect gene expression levels of splicing factors. However, no significant change was measured for 227 genes encoding splicing regulators (Supplementary Data 1, Fig. 2a), thereby suggesting a direct role of Tax in alternative splicing regulatory mechanisms. To tackle this question, we focused on the auxiliary component of the spliceosome DDX17, which has been previously identified, but not validated, in a recent mass spectrometry screen for putative protein partners of Tax[28].

We therefore aimed to validate the interaction between Tax and DDX17. As shown in Fig. 2b, Tax co-immunoprecipitated with the two endogenous isoforms of DDX17, namely p72 and p82. Reciprocal IP confirmed this interaction (Fig. 2c). Due to the involvement of NF-κB signaling in Tax-positive cells (Fig. 1d[11]), we examined whether DDX17 interacts with a Tax mutated form, namely M22 (G137A, L138S), which is defective for IKK and NF-κB activation[29–32]. Despite similar expression levels and immunoprecipitation efficiencies of Tax and M22 (Fig. 2d), we failed to detect any interaction between M22 and DDX17 (Fig. 2b,c), suggesting that NF-κB is required for recruiting DDX17. In this setting, RELA co-immunoprecipitated with DDX17 and Tax, but not with M22 (Fig. 2b, c). Moreover, DDX17 was co-immunoprecipitated with RELA in a Tax-dependent manner (Fig. 2e). This interaction did not require RNA as the DDX17:RELA complex remained detected when cell extracts were pretreated with RNAse A (Fig. 2f).

As DDX17:RELA complexes were observed neither in control cells (that do not expressed Tax) nor in M22-expressing cells, this suggested that NF-κB activation is necessary for the binding of DDX17 to RELA. This hypothesis was confirmed by exposing M22-expressing cells to TNFα, a potent NF-κB activator that allowed retrieving DDX17:RELA complexes (Fig. 2g). Altogether,

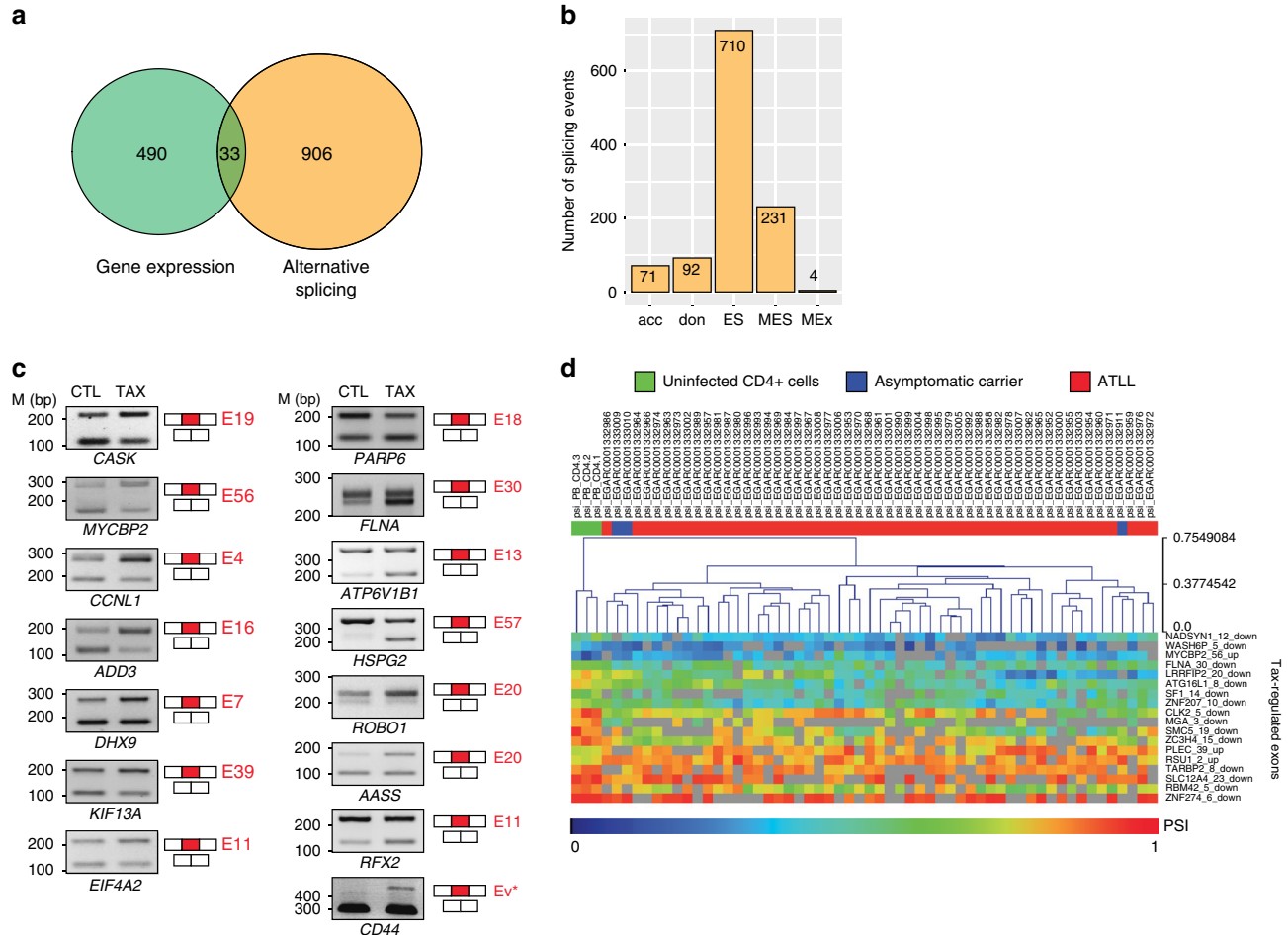

**Fig. 1 Tax induces alternative splicing modifications independently of its transcriptional effects. a** Genes regulated at the steady-state expression level and at the splicing level upon Tax expression in HEK cells. The significance thresholds were typically set to 10% for ΔPSI (differential percentage of spliced-in sequence) and 0.6 for log2-gene expression changes ($p < 0.05$, Fisher's exact test), respectively. **b** Different alternative splicing events induced by Tax: alternative acceptor (acc), alternative donor (don), exon skipping (ES), multi-exon skipping (MES), and multi-exclusive exon skipping (MEx). **c** Validation of alternative splicing predictions by RT-PCR (using 35 cycles). The exon number is indicated in red. CD44 full variants (Ev*) were assessed using primers C13 and C12A (Supplementary Fig. 5). Representative image from three independent experiments is shown. **d** Exon-based hierarchical clustering. Kruskal–Wallis ANOVA ($p < 0.05$) was carried out with Mev4.0 software (http://www.tm4.org/) using the PSI values of exons that share similar regulations upon Tax and in clinical samples (EGAS00001001296). Only the most significant exon regulations are presented. Source data are provided as a Source Data file.

these results revealed that Tax dynamically orchestrates interactions between the transcription factor RELA and the splicing regulator DDX17 by activating the NF-κB signaling pathway (Fig. 2h).

**Tax-mediated effects on splicing depend on DDX5/17.** To estimate the role of DDX17 in Tax-regulated splicing events, RNA-sequencing was performed using 293T-LTR-GFP cells expressing or not Tax and depleted or not for DDX17 and its paralog DDX5, which cross-regulate and complement each other[22,33,34]. Tax had no effect on the expression of DDX5 and DDX17 (Figs. 2a and 3a) and RELA protein level was not significantly changed upon both Tax expression and DDX5/17 silencing (Fig. 3b).

Overall, 58.5% (648/1108) of Tax-regulated exons were affected by DDX5/17 knockdown, a significantly higher proportion than expected by chance (Fig. 3c, Supplementary Fig. 2a). Of particular significance, 423 Tax-induced splicing events were completely dependent on the presence of DDX5/17 (Supplementary Data 3). For example, DDX5/17 silencing completely abolished the

Tax-mediated effect on splicing of SEC31B, CASK, MYCBP2, CCNL1, ROBO1, and ADD3 transcripts (Fig. 3d). Tax, as well as siRNA-mediated DDX5/17 depletion, had no marked effects on gene expression levels of those genes (Supplementary Fig. 2c, d). Of note, splicing specific RT-PCR assays permitted to validate the effect of DDX5/17 on Tax-dependent splicing changes for CD44, ADD3, and EIF4A2 transcripts, even though their predicted differential inclusion fell below the arbitrary computational threshold (Fig. 3d and Supplementary Fig. 2b). This suggested that the contribution of DDX5/17 to Tax-mediated alternative splicing regulation might be under-estimated.

As NF-κB activation modified the interactions between DDX17, RELA, and Tax (Fig. 2), we next examined the interplay between NF-κB activation and DDX17-mediated splicing regulation. As shown in Fig. 3d, M22 did not have any effect on DDX5/17-sensitive splicing events, arguing that Tax splicing targets are regulated by RNA helicases DDX5/17 in an NF-κB-dependent manner. This was further confirmed using siRNA-mediated depletion of RELA that abolished DDX17-dependent splicing regulations by Tax (Fig. 3d).

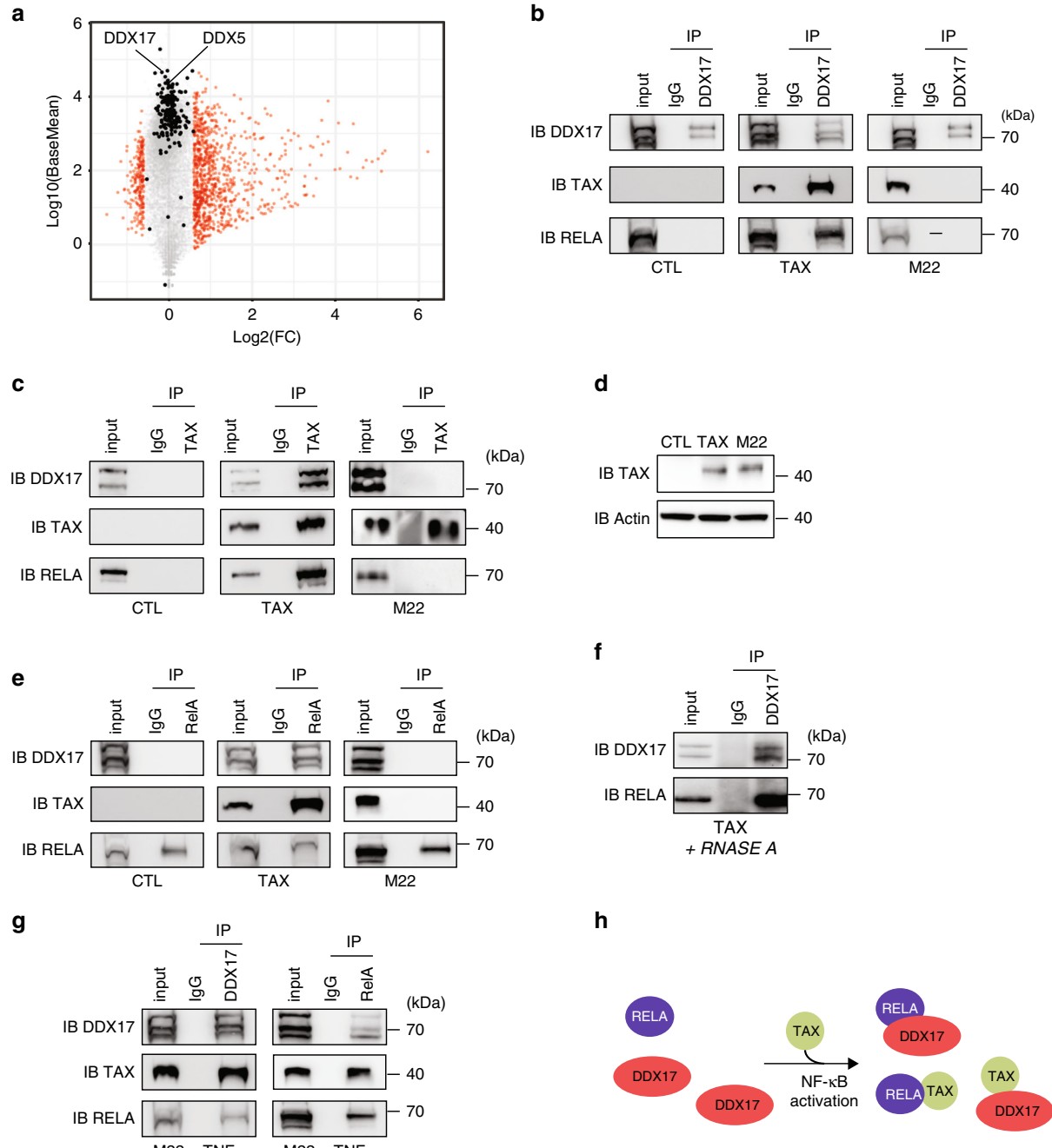

**Fig. 2 Physical interactions between Tax, RELA, and DDX17 in an NF-kB dependent manner. a** Mean average plot ($n = 3$, $p < 0.05$) of cellular gene expressions upon Tax. Each gene is plotted according to its expression level (log10(BaseMean) from DESeq2 analysis) and to fold change (log2-FC) upon Tax. Red dots show significant gene expression changes in HEK cells (log2-FC > 0.6, $p < 0.05$, Fisher's exact test). Black dots highlight genes encoding splicing factors. DDX5 and DDX17 are indicated. **b** Immunoprecipitation assays (IP) were carried out in HEK cells using isotype IgG or anti-DDX17 (**b, g**), anti-RELA (**e, g**), and anti-Tax (**c, g**) antibodies, followed by immunoblotting (IB) with indicated antibodies. **d** Western blot analysis of Tax and M22 expression 48-h post-transfection. **f** RNA-free IP assays. **g** TNFa exposure of M22-expressing cells promotes RELA–DDX17 interactions. **h** Model of NF-κB-dependent interplay between Tax, RELA, and DDX17. For **b**–**g**, a representative image from three independent experiments is shown. Source data are provided as a Source Data file.

**Functional insight of Tax and DDX17-dependent spliced genes**. Gene ontology (GO) analysis of quantitatively altered genes revealed several signaling pathways that are well described in Tax-expressing cells, including NF-κB, TNF, and MAPK signaling (Supplementary Fig. 3C)[11,35]. In contrast, genes modified at the splicing level belong to membrane-related regulatory processes including focal adhesion and ABC transporters (Supplementary Fig. 3C). In particular, the term *Focal Adhesion* was

shared between Tax splicing targets identified in infected samples (RNA-seq datasets, Supplementary Data 1) and those regulated by DDX17 (Supplementary Data 3) (Supplementary Fig. 3C). In addition, using an exon ontology (EO) approach that we recently developed to estimate enrichment in protein features encoded by exons[36], we found that Tax- and DDX17-regulated exons encoded for regions involved in functionally validated post-translational modifications (PTM), protein structure and

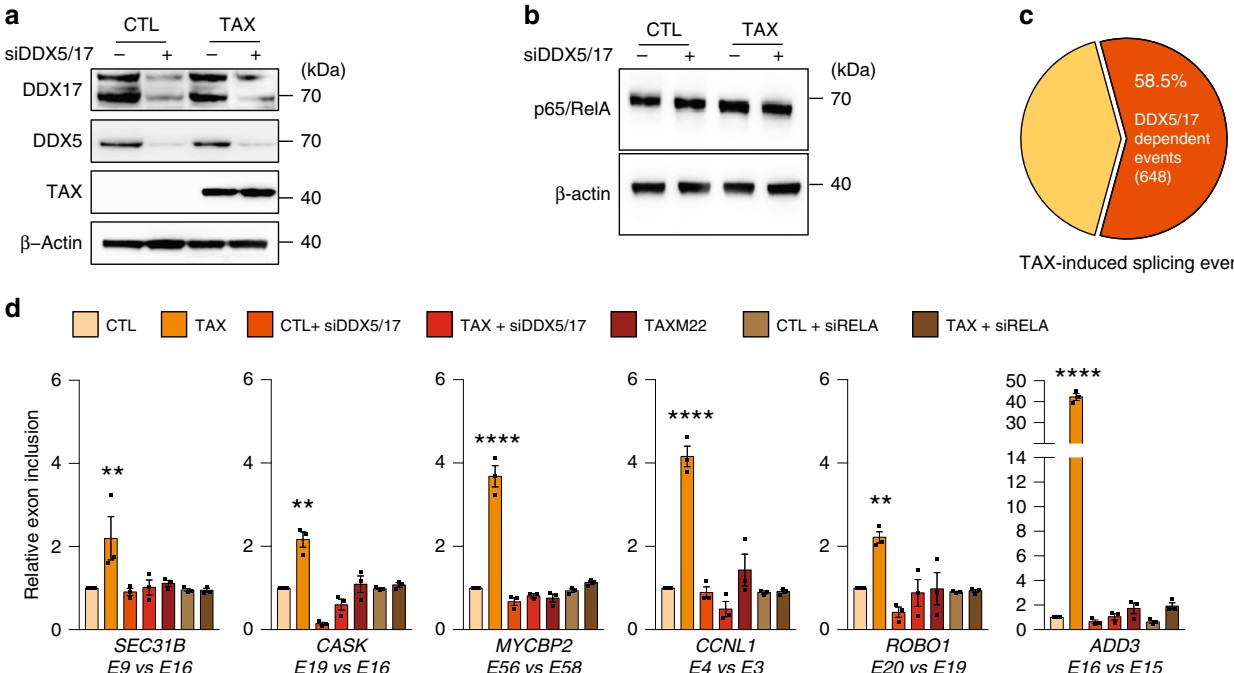

**Fig. 3 DDX5/17 regulates Tax splicing targets in an NF-kB-dependent manner. a** Western blot analysis of DDX5 and DDX17 expression in HEK cells expressing or not Tax and depleted or not of DDX5 and DDX17 by siRNA. **b** Western blot analysis of RELA and β-actin upon Tax expression and siRNA-DDX5/17 delivery. **c** Splicing events modified upon the depletion of DDX5/17 in Tax-positive HEK cells. The significant threshold was set to ≥2 in comparisons between TaxvsCTL and TaxsiDDX5/17vsCTL. For **b** and **c**, a representative image from three independent experiments is shown. **d** Validation of alternative splicing predictions of a set of Tax- and DDX5/17-regulated exons. TaxM22 and siRNA-mediated RELA depletion were used in order to assess the dependency of splicing events on NF-κB activation. Histograms represent the results of exon-specific quantitative RT-PCR measurements computed as a relative exon inclusion (alternatively spliced exon vs constitutive exon reflecting the total gene expression level). All genes but MYCBP2 were unmodified at the whole transcript level upon Tax expression (Supplementary Fig. 2c). Data are presented as the mean ± SEM values from biological replicates. Each black square represents a biological replicate. Statistical significance was determined with two-way ANOVA followed by Fisher's LSD test (**$p < 0.01$, ****$p < 0.0001$). Exact $p$-values for Tax vs CTL: 0.0068 for SEC31B; 0.0084 for CASK; <0.0001 for MYCBP2; <0.0001 for CCNL1; 0.0063 for ROBO1; <0.0001 for ADD3. Source data are provided as a Source Data file.

binding functions (Supplementary Data 4 and Supplementary Fig. 3C). These data are reminiscent of PTM related to protein regions encoded by variable CD44 exons that also control the cell-adhesion properties of CD44 (Supplementary Fig. 3D). Accordingly, we observed that Tax-expressing HEK cells displayed switched cell-adhesion properties from hyaluronate- to type IV collagen-coated surfaces, which is in accordance with the substrate affinity of the CD44 v10 isoform[37] (Supplementary Fig. 3E).

**RELA recruits DDX17 at the vicinity of genomic exons**. The results described above prompted us to focus on CD44 as gene model to further examine the mechanisms underlying RELA and DDX17 splicing regulations upon Tax. CD44 is composed of 10 constitutive exons and 10 variable exons. The constitutive exons 1–5 and 15–20 encode the standard CD44 transcripts, while CD44 variants (CD44v) are produced by extensive splicing leading to alternative inclusion of variable exons 5a-14 also named v1-v10 (Fig. 4b)[38]. As shown in Fig. 4a, the exon v10 inclusion rate is markedly influenced by Tax in a DDX5/17- and NF-κB activation-dependent manner. The importance of NF-κB in this process was further confirmed as the inactivation of NF-κB via the ectopic expression of the IκBα super repressor (IκBSR) abolished the effects of Tax on CD44 v10 inclusion (Supplementary Fig. 4A). Remarkably, in line with previous work suggesting that NF-κB directly regulates the CD44 promoter, we also noticed a slight reduction in CD44 expression in Tax-expressing cells knocked down for RELA (Supplementary Fig. 4B), thereby defining CD44

regulations as an appropriate situation to address whether or not RELA-dependent transcription and splicing are two interrelated processes.

Using quantitative ChIP (qChIP) analyses, we first observed that Tax expression led to recruit RELA not only to the CD44 promoter, but also to a genomic region spanning the alternative exon v10, but not a downstream constitutive exon (E16) (Fig. 4b, c, left panel). To assess whether RELA occupancies at the v10 exon and CD44 promoter are interrelated, a stable cell line was generated in which the κB site localized at –218 bp from the transcription start site (TSS) was deleted using a CRISPR-Cas9 approach. Positive clones (CD44ΔkB) were screened and sequenced to confirm the 40 bp deletion in the promoter region (Fig. 4b). As expected, Tax expression failed to promote RELA binding at the promoter in CD44ΔkB cells (Fig. 4c, right panel), coinciding with a decreased CD44 expression comparable to that observed upon RELA depletion (Supplementary Fig. 4B, C). Nevertheless, Tax still promoted RELA binding at the v10 region. Importantly, Tax expression induced v10 inclusion at a similar level in both CD44ΔkB and parental cells (Fig. 4d). These results suggested that Tax-mediated effect on exon v10 splicing could depend on RELA binding in the vicinity of the alternative v10 exon. Supporting this hypothesis, the analysis of publicly available RELA ChIP-seq datasets revealed that intragenic RELA peaks are significantly closer to alternative exons than to constitutive exons (Supplementary Fig. 4D). More particularly, alternative exons regulated by Tax, including those sensitive to DDX5/17 depletion, were identified in genomic regions enriched in NF-κB-binding sites compared to alternative exons non-regulated by Tax (Fig. 4e).

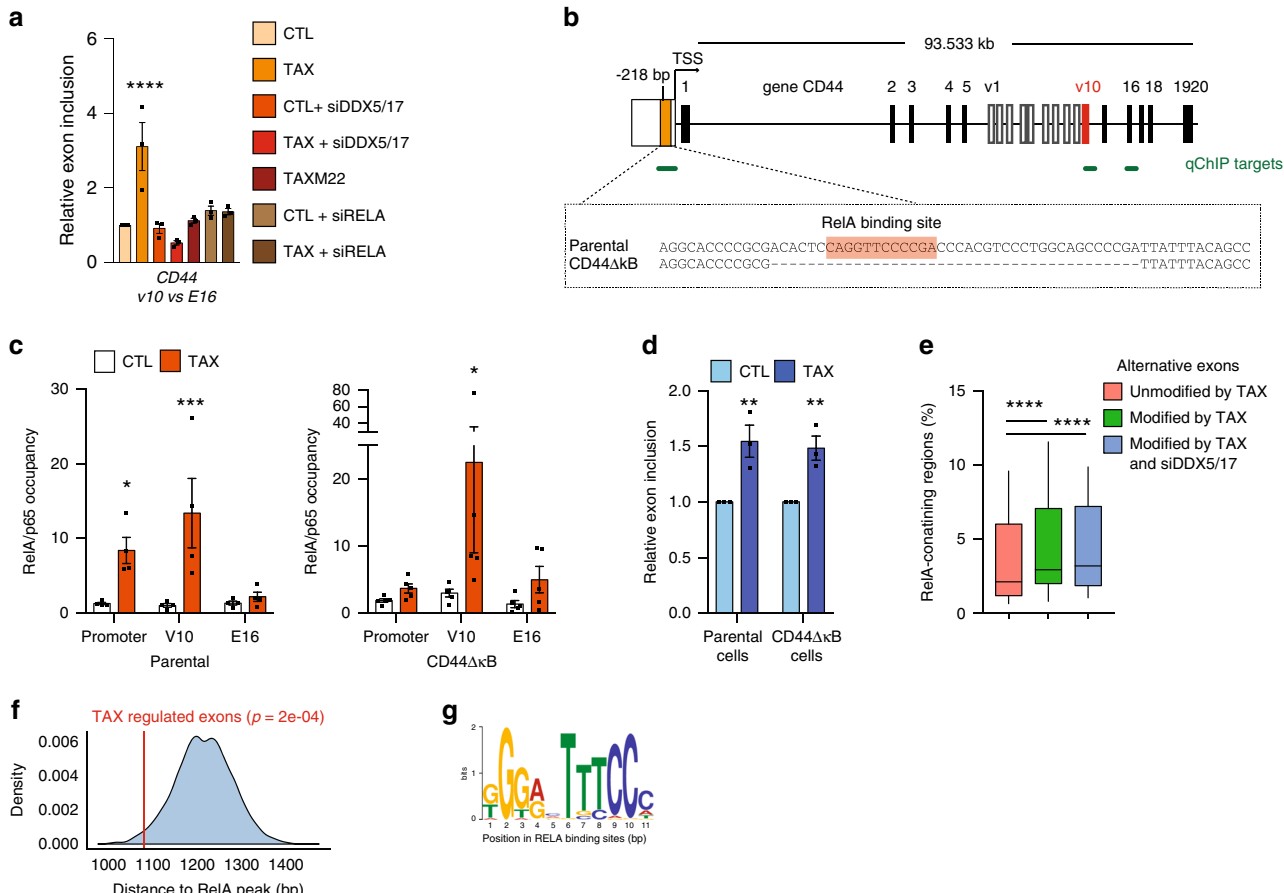

**Fig. 4 Intragenic RELA-binding sites associate with alternative splicing events. a** Alternative splicing modifications of the *CD44* exon v10 upon in HEK cells expressing or not Tax and knocked down or not for RELA or DDX5/17 expressions. TaxM22 and siRNA-mediated RELA depletion were used to assess the dependency of splicing events after NF-κB activation. Histograms represent the results of exon-specific quantitative RT-PCR measurements computed as a relative exon inclusion (alternatively spliced exon vs constitutive exon reflecting the total gene expression level). **b** Schematic representation of the human *CD44* gene. Black and white boxes represent constitutive and alternative exons, respectively, as previously annotated (50). The orange box represents the kB site localized at −218 bp from the TSS and the 40 bp fragment deleted by CRISPR/Cas9 in CD44ΔkB HEK cells. **c** qChIP analysis of RELA occupancy across the promoter, the exon v10, and the constitutive exon E16 of CD44. RELA enrichment is expressed as the fold-increase in signal relative to the background signal obtained using a control IgG. **d** Relative exon inclusion of CD44 exon v10 was quantified by qRT-PCR in parental cells and its CD44ΔkB counterparts. **e** Distribution of alternative exons that are regulated or not by Tax and DDX5/17 in RELA-enriched intragenic regions. The analysis was restricted to alternative exons expressed in HEK cells and regulated or not by Tax. Boxes extend from the 25th to 75th percentiles, the mid line represents the median and the whiskers indicate the maximum and the minimum values. **f** Bootstrapped distribution of median distance between intragenic RELA peaks and either Tax-regulated exons (red line, 1079 bp) or randomly chosen exons ($10^5$ repetitions) (blue). *p*-values were determined by sample *t*-test. **g** Consensus de novo motif for RELA-binding sites <1 kb of Tax-regulated exons. Data are presented as the mean ± SEM values from biological replicates. Each black square represents a biological replicate. Statistical significance was determined with two-way ANOVA followed by Fisher's LSD test (**p* < 0.05, ***p* < 0.01, ****p* < 0.001) (**a**, **c**, and **d**) and two-tailed Wilcoxon test (**e**, *****p* < 0.0001). Exact *p*-values for Tax vs CTL: **a** <0.0001, **c** parental, promoter: 0.0248 and V10: 0.0005; CD44ΔkB, V10: 0.021; **d** parental: 0.0028; CD44ΔkB: 0.0054. Source data are provided as a Source Data file.

Accordingly, we observed that RELA-binding sites are often found in the vicinity of Tax-regulated exons (Fig. 4f). Using the MEME-ChIP suite as motif discovery algorithm[39], we uncovered that RELA-binding sites located within the closest range (<1 kb) of Tax-regulated exons coincided with the typical NF-κB consensus motif (Fig. 4g). Furthermore, this subset of Tax-regulated exons displayed weak 3′ and 5′ splice sites together with significant low minimum free energy (MFE) value (Supplementary Fig. 4E) and high GC-content (Supplementary Fig. 4F) when compared to all human exons. This emphasizes the high potential of these splice sites to form stable secondary RNA structures, a typical feature of exons regulated by RNA helicases DDX5/17[33].

Taken together, these data define a signature of splicing target specificity for RELA, and they suggest that RELA and DDX17 might control together the inclusion of a subset of Tax-regulated exons. We therefore investigated the genomic occupancy of some

target exons by RELA and DDX17 by qChIP analysis of cells expressing or not Tax. For all tested genes (*CD44*, *SEC31B*, *CASK*, and *MYCBP2*), both RELA and DDX17 bound specifically the regulated alternative exon in a Tax-dependent manner, compared to a downstream constitutive exon (Fig. 5a and Supplementary 4g). Remarkably, the knockdown of RELA expression by siRNA affected DDX17 chromatin occupancy on Tax-regulated exons (Fig. 5a), despite its slight positive impact on DDX17 expression (Supplementary Fig. 2d). We further validated the RELA/DDX17-dependent alternative splicing events in C91PL and ATL2 cells compared to the non-infected MOLT4 cells (Fig. 5c). The magnitude of splicing regulation was higher in ATL2 than in C91PL cells, and appeared to positively correlate to a higher expression level of both DDX17 and RELA, and a high level of constitutive NF-κB activation, as reflected by IL8 expression (Fig. 5b). As formerly observed in HEK cells expressing Tax, these

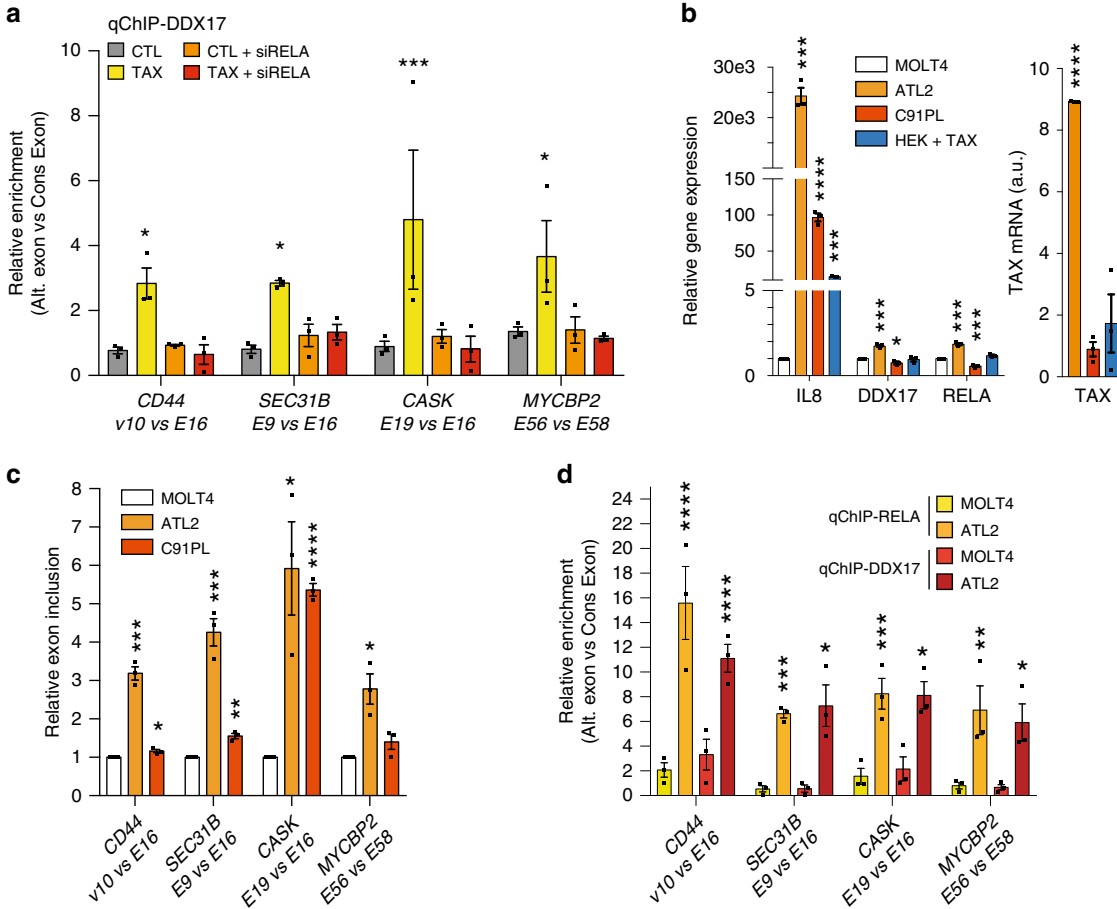

**Fig. 5 RELA locally recruits DDX17 at the genomic target exons, leading to splicing regulation. a** Relative occupancy of DDX17 at Tax-regulated genomic exons in cells that did or did not express Tax and knocked down or not with RELA-specific siRNA. **b** Relative gene expression levels of *IL8*, *DDX17*, and *RELA* in ATL2, C91PL, or HEK cells transiently transfected by pSG5M-Tax as compared uninfected MOLT4 cells. Tax mRNAs levels are expressed in arbitrary units (a.u.). **c** Alternative splicing modifications in the HTLV-1-infected cell lines ATL2 and C91PL as compared to those in the uninfected cell line MOLT4. Relative exon inclusion was measured as described in Fig. 3. **d** Relative RELA and DDX17 occupancies of regulated exons in ATL2 cells as compared to MOLT4 cells. Each occupancy of regulated exon by RELA and DDX17 is represented as a fold of that measured at its neighboring constitutive exon. Source data are provided as a Source Data file. Data are presented as the mean ± SEM values from biological replicates. Each black square represents a biological replicate. Statistical significance was determined with two-way ANOVA followed by Fisher's LSD test (**a**, **d**) and two-tailed unpaired *t*-test (**b**, **c**) (*$p < 0.05$, **$p < 0.01$, ***$p < 0.001$, ****$p < 0.0001$). Exact *p*-values for Tax vs CTL (**a**): 0.0312 (CD44), 0.0337 (SEC31B), 0.0002 (CASK), 0.0171 (MYCBP2). Exact *p*-values for ATL2- and C91PL vs MOLT4 (**b**): 0.0001, 0.0004, 0.0003 and <0.0001, 0.0365, 0.0007 corresponding to IL8, DDX17, and RELA, respectively. For Tax expression, $p = 0.0016$ HEK + Tax vs ATL2. Exact *p*-values for ATL2- and C91PL vs MOLT4 (**c**): 0.0002, 0.0008, 0.0153, 0.0105 and 0.019, 0.0013, <0.0001, 0.11 for CD44, SEC31B, CASK, and MYCBP2 respectively. Exact *p*-values (**d**) for ATL2 vs MOLT4: qChIP-RELA < 0.0001 (CD44), 0.0003 (SEC31B), 0.0004 (CASK), 0.0029 (MYCBP2); qChIP-DDX17 0.0004 (CD44), 0.0211 (SEC31B), 0.0207 (CASK), 0.0184 (MYCBP2). Source data are provided as a Source Data file.

splicing events in HTLV-1-infected cells coincided with a high and significant increase in chromatin occupancy of both RELA and DDX17 at the vicinity of regulated exons (Fig. 5d). These data reveal that the RELA:DDX17 axis in the splicing regulatory network pertains to both HEK293T and CD4+-infected cells expressing Tax.

**Causal relationship linking RELA and DDX17 to splicing regulation**. To more confidently assess the causative relationship linking RELA and DDX17 to alternative splicing, we experimentally tethered DDX17 or RELA to the *CD44* v10 exon locus using modified TALE (Transcription-Activator-Like-Effector)[40]. For this, we designed a TALE domain that recognizes specifically an exonic 20 bp DNA sequence located 12 bp upstream from the 5′ splice site (SS) of exon v10. This TALE domain was fused to either RELA or DDX17 proteins. We also used an additional construct consisting in the same TALE fused to GFP to rule out

non-specific effects resulting from the DNA binding of the TALE. After transient transfection of each TALE construct into 293T-LTR-GFP cells (Fig. 6a), we monitored the relative effects on the recruitment of endogenous RELA and DDX17 as well as on exon v10 splicing; all results were normalized and expressed as relative effects compared to the TALE-GFP (Fig. 6a–c). As expected, and validating our approach, TALE-RELA tethering to the exon v10 led to a significant chromatin recruitment of RELA to its target site, and not to the downstream exon E16 used as control (Fig. 6a, left panel). Further, a significant and specific DDX17 enrichment was observed at exon v10 after expression of TALE-RELA as compared to TALE-GFP (Fig. 6a, left panel), indicating that tethering RELA to exon v10 induced local recruitment of the endogenous DDX17 protein. At the RNA level, this TALE-RELA-mediated recruitment of DDX17 coincided with a significant increase in the exon v10 inclusion rate (Fig. 6a, right panel).

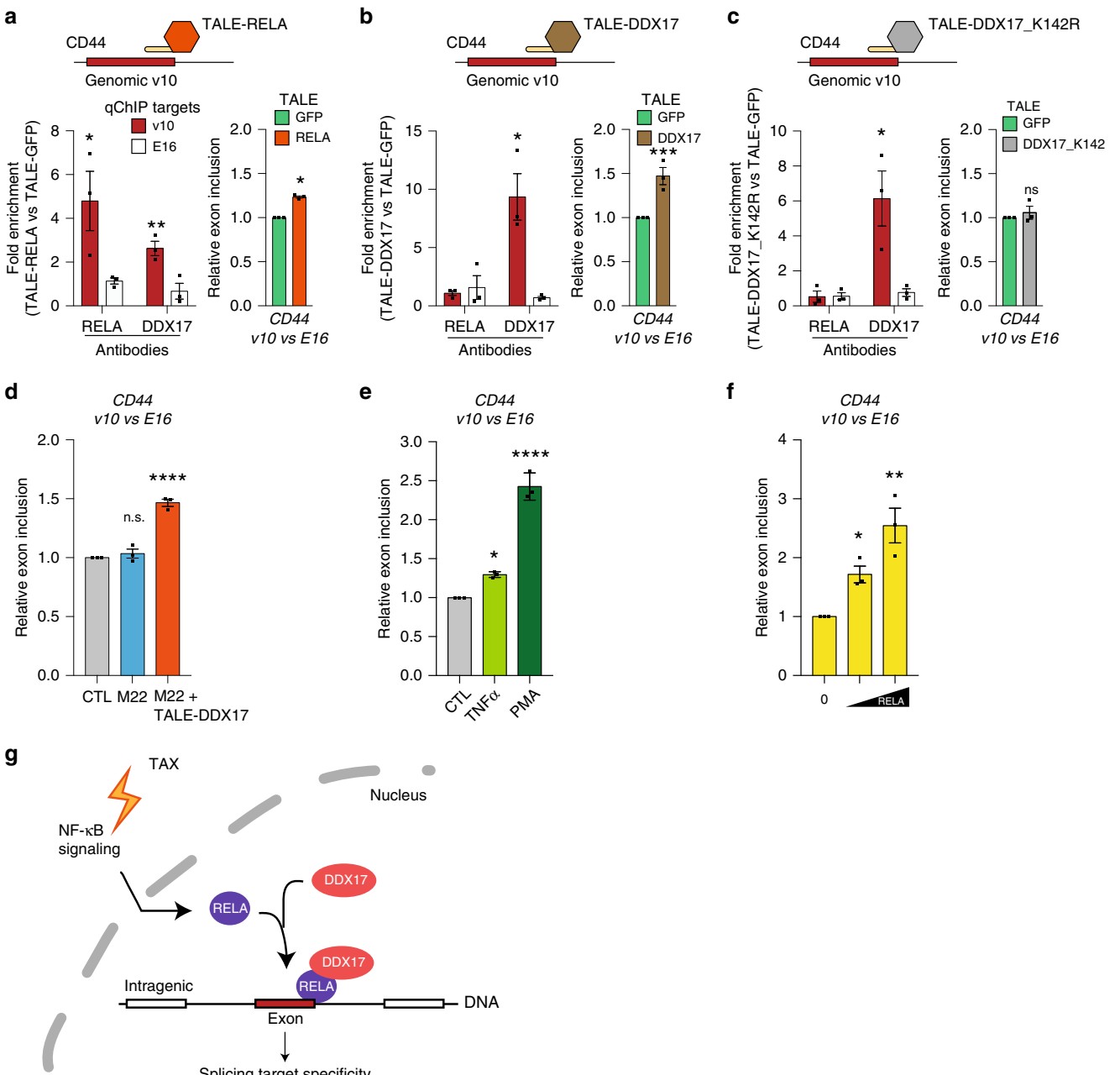

**Fig. 6 Chromatin relationship between RELA and DDX17. a–c** Chromatin and splicing regulation upon TALE-mediated tethering of RELA and DDX17. The TALE domain was designed to bind the v10 exon of *CD44* and fused to either GFP (**a–c**), RELA (**a**), DDX17, (**b**), or a helicase-deficient mutant DDX17_K142R (**c**). The effect of TALEs on RELA and DDX17 chromatin enrichment (left panels) and on the relative v10 exon inclusion (right panels) was monitored in HEK cells by qChIP and qRT-PCR, respectively. Results were normalized to measures obtained in TALE-GFP assays. (**d**) Relative exon inclusion rate of exon v10 of CD44 in HEK cells expressing or not the Tax mutant M22 and the TALE-DDX17 construct. **e** Relative exon inclusion rate of exon v10 of *CD44* in HEK cells exposed to TNFα and PMA. **f** Relative exon inclusion rate of exon v10 of *CD44* in HEK cells transiently transfected with increasing amounts of RELA expression vector (200 and 500 ng). **g** Model of NF-κB-dependent regulation of alternative splicing. Upon NF-κB activation, DNA-bound RELA proteins act as chromatin anchors for DDX17, which then provides splicing target specificity due to its RNA helicase activity. Data are presented as the mean ± SEM values from biological replicates. Each black square represents a biological replicate. Statistical significance was determined with two-tailed unpaired *t*-test (qChIP in **a–c**) and one-way ANOVA followed by Fisher's LSD test (relative exon inclusion (REI) in **a–f**) (*$p < 0.05$, **$p < 0.01$, ***$p < 0.001$, ****$p < 0.0001$). In **a–c**, exact *p*-values for TALE-RELA vs TALE-GFP on V10: 0.049 (qChiP RELA), 0.0079 (qChIP DDX17), 0.0276 (REI). Exact *p*-values for TALE-DDX17 vs TALE-GFP on V10: 0.0139 (qChIP DDX17), 0.0276 (REI). Exact *p*-values for TALE-DDX17_K142R vs TALE-GFP on V10: 0.0312 (qChIP DDX17). In **d–f**, exact *p*-values are <0.0001 (M22 + TALE-DDX17 (**d**)), 0.0125 (TNFa (**e**)), <0.0001 (PMA (**e**)), 0.0369 (0.2 μg (**f**)), 0.0012 (0.5 μg (**f**)). Source data are provided as a Source Data file.

We next investigated whether DDX17 tethering could result in similar effects. Quantitative ChIP analysis demonstrated that DDX17 was properly tethered to exon v10 when fused to the designed TALE, but TALE-DDX17 had no effect on RELA

recruitment (Fig. 6b, left panel). This was expected since the formation of RELA:DDX17 complexes only occurs upon NF-κB activation (Fig. 2). Nevertheless, TALE-DDX17–expressing cells exhibited a reproducible and significant increase in v10 inclusion

(Fig. 6b, right panel), indicating that chromatin-bound DDX17 alone (e.g., without RELA) can modulate splicing efficiency. Of note, the levels of v10 exon inclusion induced by TALE-RELA and TALE-DDX17 were comparable to that measured in cells transiently transfected with a Tax expression vector (Fig. 4d and Supplementary Fig. 4A). Although less quantitative, a nested RT-PCR assay clearly confirmed these results (Supplementary Fig. 5b). Strikingly, however, the mutant TALE-DDX17_K142R (a DDX17 helicase mutant[33,41–43]) failed to increase the levels of exon v10 inclusion, despite a clear chromatin enrichment of the DDX17 mutant (Fig. 6c and Supplementary Fig. 5b). Collectively, these results demonstrate that RELA binding in the vicinity of genomic exons recruits the RNA helicase DDX17, which positively regulates the inclusion rate of the target exon due to its RNA helicase activity. Of note, this causal relationship between RELA, DDX17, and splicing regulation was observed irrespectively of the expression of Tax. Indeed, TALE-DDX17 rescued the inclusion of exon v10 in cells expressing the NF-κB defective mutant M22 (Fig. 6d). In addition, Tax-independent activation of NF-κB in HEK cells (using either TNFα or Phorbol 12-myristate 13-acetate (PMA)) mimicked the effects of Tax on splicing regulation (Fig. 6e). More importantly, ectopic RELA expression had similar effects, in a dose-dependent manner (Fig. 6f). Further, we observed a dynamic induction of RELA:DDX17 interactions after RELA expression or PMA activation (Supplementary Fig. 5c), but not after TNFα activation, likely due to a lower level of RELA nuclear translocation in this condition (Supplementary Fig. 5d). Collectively, these results suggest that the mechanistic role of Tax is to promote DDX17-dependent splicing regulation by promoting constitutive activation of NF-κB pathway.

## Discussion

Since the finding of splicing dysregulations in HTLV-1-infected individuals[4,24,27,44], deciphering how HTLV-1 interferes with the splicing regulatory network has become a new challenging issue for improving our knowledge of HTLV-1 infection and its associated diseases. Here, we provide the molecular evidence that upon Tax-induced NF-κB activation, RELA directly regulates splicing by binding to gene bodies at the vicinity of GC-rich exons and by locally recruiting the splicing factor DDX17, which regulates splicing via its RNA helicase activity.

Our results demonstrate that Tax deeply impacts alternative splicing independently from its effects on transcription. In addition, Tax-regulated exons were found in transcripts enriched in functional pathways that are distinct from those enriched by Tax transcriptional targets, suggesting that splicing reprogramming may constitute an additional layer of regulation by which HTLV-1 modifies the host cell phenotype. Supporting this, we also showed that the Tax-induced splicing variant *CD44 v10*, which was previously identified in circulating blood of HAM/TSP patients[27] and confirmed here ex vivo in infected CD4+ T-cell clones, contributes to modulating cell-adhesion affinity in vitro. GO analyses of Tax splicing targets also identified the GO terms phosphatidylinositol signaling system and inositol phosphate metabolism, two processes that are particularly connected to NF-κB signaling and that have critical roles in oncogenesis and disease progression of malignant diseases, including ATLL[45,46]. Furthermore, exon ontology analysis predicted that critical changes at the protein level would affect the experimentally validated protein structure and post-translational modifications, which together are likely to affect the connectivity network between proteins, and subsequently contribute to modifying cell phenotypes. This suggests that, beside its transcriptional effects, splicing regulatory functions of Tax might contribute to its oncogenic properties. Indeed, a large number of Tax-regulated

exons could be observed in ATLL samples, which rarely express Tax but typically exhibit NF-κB addiction for survival and proliferation[24,26,47].

At the molecular level, we showed that increased chromatin occupancy of RELA upon Tax expression is not restricted to promoter regions but also occurs in the vicinity of exons that are regulated at the splicing level (Figs. 4 and 5). Exons regulated by Tax, especially those localized within 1 kb of intragenic RELA-binding sites, are characterized by a high GC-content, a typical feature of exons regulated by the DDX5 and DDX17 RNA helicases[33] (Supplementary Fig. 4E, F). Accordingly, we found that a majority of Tax-regulated exons depend on the expression of these proteins (Fig. 3c). A local chromatin recruitment of DDX17 and RELA was validated on several Tax-regulated exons (Figs. 4 and 5). More importantly, we identified a confident causal relationship between exon tethering of RELA, local recruitment to chromatin of DDX17, and subsequent splicing regulation via DDX17 RNA helicase activity (Fig. 6). This catalytic activity of DDX17 was strictly required for its splicing regulatory functions (Fig. 6), as previously reported[33]. Indeed, the RNA helicase activities of DDX5 and DDX17 have been implicated in resolving RNA structures, facilitating the recognition of the 5′ splice site (which can be embedded in secondary structures), and exposing RNA-binding motifs to additional splicing regulators[33,42,48–50]. However, even though some RNA-binding specificity has been reported for DDX17[51,52], these RNA helicases are devoid of a proper RNA-binding domain, and their activity in splicing likely depends on additional factors that are able to provide target specificity. Here, we suggest that RELA is a DDX17 recruiter, acting as a chromatin anchor for DDX17 in the vicinity of exons dynamically selected upon NF-κB activation. While performing complementary experiments, we notified that siRNA-DDX5/17 affected the RELA chromatin occupancy of both exons and promoters regulated by Tax (Supplementary Fig. 6). This is reminiscent of recent report indicating that DDX5/17 depletion affects interactions between the transcription factor REST and DNA, defining DDX5/17 as REST transcriptional coregulators[22]. Altogether, these data indicate that the RELA–DDX17 chromatin interplay relies on complex mechanisms that deserve future investigations.

The target specificity of NF-κB factors remains a complex question. It has been estimated that ~30–50% of genomic RELA-binding sites do not harbor a typical NF-κB site, and only a minority of RELA-binding events have been associated with transcriptional change[16–19], thereby indicating that neither having a consensus site nor significant NF-κB occupancy are sufficient criteria for defining RELA's target specificity. Here, we identified a typical NF-κB consensus motif at RELA-binding loci that are close to alternatively spliced exons but we also uncovered that weak splice sites, low MFE, and significant GC-content bias of exons likely contribute to RELA's target specificity. Because low MFE and high GC-content confer a high propensity to form stable RNA secondary structures, the recognition and the selection of such GC-rich exons with weak splice sites by the splicing machinery typically depend on the RNA helicases DDX5/17[33]. Based on these observations, we propose a model of RELA-induced splicing target specificity (Fig. 6g), whereby, upon NF-κB activation, RELA binds to intragenic-binding consensus motifs and locally recruits DDX17. When the RELA:DDX17 complex is located in close proximity of GC-rich exons flanked by weak splice sites, DDX17 can increase their inclusion rate by unwinding GC-rich secondary structures of the nascent RNA transcript, and potentially also by unmasking binding motifs for additional splicing regulators. Of note, although Tax was identified in both RELA- and DDX17-containing complexes, we demonstrated that Tax-independent NF-κB activation is sufficient

for promoting interactions between RELA and DDX17 and the corresponding splicing regulation. Without ruling out possible additional effects of Tax on the RELA:DDX17 complex, such as favoring dimer formation (as already proposed for NF-κB dimers[15]), these data indicate that Tax exacerbates a dynamic and physiologic process involving RELA in splicing target specificity.

In conclusion, our results provide conceptual advances for understanding how cell signaling pathways may drive target specificity in splicing by dynamically recruiting cognate transcription factors at the vicinity of target exons that act as chromatin anchor for splicing regulators. In the context of NF-κB signaling, such a mechanism likely has a significant impact on cell fate determination and disease development associated with HTLV-1 infection and other situations linked to chronic NF-κB activation, such as human inflammatory diseases and cancer.

## Methods

**Cell culture and transfections**. Peripheral blood mononuclear cells (PBMCs) were obtained by Ficoll separation of whole blood from HTLV-1-infected individuals in the context of a Biomedical Research Program approved by the Committee for the Protection of Persons, Ile-de-France II, Paris (2012-10-04 SC). All individuals gave informed consent. PBMCs were cloned by limiting dilution in RPMI 1640 supplemented with penicillin and streptomycin, sodium pyruvate, non-essential amino acids, 2-mercaptoethanol, 10% filtered human AB serum, 100 U/ml recombinant IL-2 (Chiron Corporation), and 75 μM HTLV-1 integrase inhibitor L-731,988. Clones were phenotyped by flow cytometry using antibodies against CD4 (Dako-Cytomation) and isotype-matched controls on a FACScan system using CellQuest software (Becton Dickinson). HTLV-1-positive clones were assessed by PCR. The human embryonic kidney 293T-LTR-GFP cells[53], which contain an integrated GFP reporter gene under the control of the Tax-responsive HTLV-1 LTR, were cultured in DMEM + glutamax medium supplemented with 10% heat-inactivated FBS and 1% penicillin/streptomycin. This cell line was used to measure transfection efficiency in Tax and TaxM22 conditions. In standard transfection experiments, siRNAs (Supplementary Data 5) and/or expression vectors (pSG5M empty, pSG5M-Tax-WT, pSG5M-M22) were mixed with JetPrime (Polyplus Transfection) following the manufacturer's instructions, and cells were harvested 48 h after transfection. TNFα exposure consisted in treating cells with 10 ng/ml of TNFα for 24 h. HTLV-1-chronically infected lymphocytes ATL2 (kind gift from Masao Matsuoka (Kyoto University, Kyoto, Japan) and Roberto Accolla (Università degli Studi dell'Insubria, Varese, Italia)), C91PL (kind gift from Cynthia Pise-Masison (National Cancer Institute, NIH, Bethesda, MD)) and Renaud Mahieux (Center for Research in Infectious Diseases, Lyon, France) and non-infected MOLT4 cells (CRL-1582, ATCC) were grown in RPMI 1640 medium (Gibco, Life Technologies) supplemented with 10% heat-inactivated fetal calf serum, 20 IU/ml penicillin, 20 μg/ml streptomycin, and 25 mM HEPES.

**Cell-adhesion assays**. Culture plates were prepared by coating with 40 μg/ml hyaluronic acid from human umbilical cord (Sigma) and 25 μg/ml type IV collagen from human placenta (Sigma) overnight at 4 °C. Non-specific binding sites were blocked for 1 h with PBS containing 1 mg/ml heat-denatured BSA. After three washes with 1× PBS, 5 × 10^4 cells transiently transfected with pSG5M-Tax vector or its empty control were added at 48-h post-transfection. Cell adhesion was allowed to proceed for 20 min at room temperature. Non-adherent cells were removed with three washes with 1× PBS, and adherent cells were quantified. All experiments were done in triplicate.

**Cell fractionation**. HEK cells were resuspended in A buffer complemented with DTT, PhosSTOP™ and protease inhibitor cocktail (10 mM HEPES, 10 mM KCl, 1,5 mM MgCl₂, 0,5 mM DTT, 1× PhosSTOP™, 1× protease inhibitor cocktail) then incubated on ice for 10 min. The lysates were then centrifuged at 4 °C for 3 min at 1000×g. Supernatants were next resuspended in A* buffer complemented with DTT, PhosSTOP™ and protease inhibitor cocktail (10 mM HEPES, 10 mM KCl, 1.5 mM MgCl₂, 0,2% IGEPAL® CA-630, 0.5 mM DTT, 1X PhosSTOP™, 1× protease inhibitor cocktail) then incubated on ice for 2 min. Lysates were then centrifuged at 4 °C for 1 min at 1000×g and supernatant were kept as cytoplasmic extract. Pellets were extensively washed with A* buffer and centrifugated, then resuspended in Lysis Buffer complemented with DTT, PhosSTOP™ and protease inhibitor cocktail (50 mM Tris-HCl pH 8.0, 400 mM NaCl, 5 mM EDTA, 0.2% SDS,1% IGEPAL® CA-630, 1 mM DTT, 1× PhosSTOP™, 1× cOmplete™ EDTA free protease inhibitor cocktail (Roche)) and incubated 30 min one ice before sonication using Diagenode Bioruptor® Plus (6 cycles 30″/30″, high power). Nuclei lysates were centrifuged at 4 °C for 10 min at 16,000×g and the supernatant were kept as nuclear extracts.

**Western blot**. Cells were washed twice with 1× PBS and total proteins were directly extracted in RIPA buffer (50 mM Tris-HCl pH 7.4, 50 mM NaCl, 2 mM EDTA, 0.1% SDS, 1× PhosSTOP™, 1× cOmplete™ EDTA free protease inhibitor cocktail (Roche)). A total of 20 μg of whole-cell proteins were separated on a NuPAGE™ 4–12% Bis-Tris Protein Gels and transferred on a nitrocellulose membrane using Trans-Blot® Turbo™ Blotting System. Membranes were saturated with 5% milk and incubated overnight at 4 °C with the primary antibodies against RELA (sc-109, Santa Cruz, 1:1000), Tax (1A3, Covalab, 1:500), DDX17 (ab24601, Abcam, 1:2000), DDX5 (ab10261, Abcam, 1:2000), actin (sc-1616, Santa Cruz, 1:1000), GAPDH (sc-32233, Santa Cruz, 1:10,000), H3 (ab1791, Abcam, 1:10,000), V5 (AB3792, Millipore, 1:1000), and α-Tubulin (sc-32293, Santa Cruz, 1:2000). After three washes with 1× TBS-Tween, membranes were incubated 1 h at room temperature with the secondary antibodies conjugated with the HRP enzyme and washed three times as above. Finally, the HRP substrate (GE Healthcare or Immobilon Forte (Millipore)) was applied to the membrane for 5 min, and the chemiluminescence was read on Chemidoc (BioRad).

**Co-immunoprecipitation**. Cells were harvested in IP lysis buffer (20 mM Tris-HCl pH 7.5, 150 mM NaCl, 2 mM EDTA, 1% NP-40, 10% glycerol). Extracts were incubated overnight with 5 μg of antibodies recognizing RELA (C20 sc-372, Santa Cruz), Tax (1A3, Covalab), and DDX17 (ProteinTech) in the presence of 30 μl Dynabeads® Protein A/G (ThermoFisher). Isotype IgG rabbit (Invitrogen) or mouse (Santa Cruz) was also used as negative control. The immunoprecipitated complexes were washed three times with IP lysis buffer.

**Chromatin immunoprecipitation**. A total of 10^7 cells were crosslinked with 1% formaldehyde for 10 min at room temperature. Crosslinking was quenched by addition of 0.125 M glycin. Nuclei were isolated by sonication using a Covaris S220 (2 min, Peak Power: 75; Duty Factor: 2; Cycles/burst: 200), pelleted by centrifugation at 1000×g for 5 min at 4 °C, washed once with FL buffer (5 mM HEPES pH 8.0, 85 mM KCl, 0.5% NP-40) and resuspended in 1 ml shearing buffer (10 mM Tris-HCl pH 8.0, 1 mM EDTA, 2 mM EDTA, 0.1% SDS). Chromatin was sheared in order to obtain fragments ranging from 200 to 800 bp using Covaris S220 (20 min, Peak Power: 140; Duty Factor: 5; Cycles/burst: 200). Chromatin was next immunoprecipitated overnight at 4 °C with 5 μg of antibodies, of anti-RELA (C20 sc-372, Santa Cruz), anti-DDX17 (19910-1-AP, ProteinTech), or anti-V5 (AB3792, Millipore), and 30 μl Dynabeads® Protein A/G (ThermoFisher) were added. Complexes were washed with 5 different buffers: Wash 1 (1% Trition, 0.1% NaDOC, 150 mM NaCl, 10 mM Tris-HCl pH 8), Wash 2 (1% NP-40, 1% NaDOC, 150 mM KCl, 10 mM Tris-HCl pH 8), Wash 3 (0.5% Trition, 0.1% NaDOC, 500 mM NaCl, 10 mM Tris-HCl pH 8), Wash 4 (0.5% NP-40, 0.5% NaDOC, 250 mM LiCl, 20 mM Tris-HCl pH 8, 1 mM EDTA), and Wash 5 (0.1% NP-40, 150 mM NaCl, 20 mM Tris-HCl pH 8, 1 mM EDTA). The immunoprecipitated chromatin was purified by phenol-chloroform extraction, and quantitative PCR was performed using Rotor-Gene 3000 cycler (Corbett) or LightCycler 480 II (Roche, Mannheim, Germany). Values were expressed relative to the signal obtained for the immunoprecipitation with control IgG. Primers used for ChIP experiments were designed for exon/intron junction (Supplementary Data 5). For TALE ChIP experiments, DDX17 and RelA enrichment were normalized to the signal observed with V5 antibody corresponding to TALE recruitment. The TALE-GFP condition was used as control and set to 1. Note that in TALE assays qChIP experiments and exon-specific RT-qPCR were carried out at 24-h post-transfection.

**RNA extraction, PCR, and real-time quantitative PCR**. Total RNAs were extracted using TRIzol (Invitrogen). RNAs (2.5 μg) were retro-transcribed with Maxima First Strand cDNA Synthesis Kit after treatment with dsDNase (Thermo Scientific) following the manufacturer's instructions. PCRs were performed using 7.5 ng of cDNAs with GoTaq polymerase (Promega, Madison, WI, USA). PCR products were separated by ethidium bromide–labeled agarose gel electrophoresis. Band intensity was quantified using the ImageLab software (BioRad). Quantitative PCR was then performed using 5 ng of cDNAs with SYBR® Premix Ex Taq TM II (Tli RNaseH Plus) on LightCycler 480 II. Relative levels of the target sequence were normalized to the 18 S or GAPDH gene expression (ΔCt), and controls were set to 1(ΔΔCt). The inclusion rate of alternative exons was calculated as $2^{-\Delta\Delta Ct}$ (included exon)/$2^{-\Delta\Delta Ct}$ (constitutive exon). Oligonucleotide sequences used are listed in Supplementary Data 5.

**RNA-seq and bio-informatic analysis**. RNA-seq analyses were performed with poly-A transcripts extracted from 293T-LTR-GFP cells transfected with pSG5M-Tax or pSG5M empty vectors and knocked down or not for DDX5-17. RNA-seq libraries were generated at Aros Applied Biotechnology (Aarhus, Denmark) using Stranded mRNA Sample Prep kit (Illumina) and sequenced using illumina HiSeq 2500 technology. Each sample had in average 6 × 10^7 of paired-end pairs of reads. RNA-seq data were analyzed using FaRLine, a computational program dedicated to analyzing alternative splicing with FasterDB database[23,54]. The gene expression level in each sample was calculated with HTSeq-count (v0.7.2)[55], and differential expression between conditions was computed with DESeq2 (v1.10.1) (abs(log2-FoldChange) ≥ 0.4, $p \leq 0.05$)[56]. Tax expression in the RNA-seq dataset EGAS00001001296 was examined by Kallisto[57] using the nucleotide sequence of Tax/Rex sequence (coordinates 6951-8078) from NC_001436.1 as a reference.

Ontology analysis were performed using DAVID software for gene ontology and Exon Ontology v1.5.0[36].

In silico screening of NF-κB-responsive elements in the *CD44* promoter sequence was carried out using the PROMO database (based on TRANSFAC v8.3)[58]. The MEME-ChIP suite was used to discover the regulatory motifs in the NF-κB ChIP-seq data[39].

To predict splice site strengthes, scores were computed using MaxEntScan[59] for the sequence covering both sides of the splicing site (using 3 bases into the exon and 6 bases into the intron for 5′ splice sites, and 20 bases into the intron and 3 bases into the exon for 3′ splice sites). MaxEntScan uses Maximum Entropy Models (MEMs) to compute log-odds ratios. The minimum free energy was computed from exon-intron junction sequences using RNAFold from the ViennaRNA package (v 2.4.1; http://rna.tbi.univie.ac.at/cgi-bin/RNAWebSuite/RNAfold.cgi). Analyzed sequences include 25 nucleotides within the intron and 25 nucleotides within the exon. The GC-content was calculated for exons defined in FasterDB[54].

The distribution of RELA peaks across alternative and constitutive exons, and the average distance between RELA peaks and Tax exon targets was measured using ChIP-seq datasets from GEO[60], ENCODE[61], and CISTROME[62] databases: from GEO GSE63736, GSM1239484, GSM486271, GSM486293, GSM486298, GSM486318, GSM847876, GSM847877, GSM2394419, GSM2394421, GSM2394423, from ENCODE ENCFF002CPA, ENCFF002CQB, ENCFF002CQJ, ENCFF002CQN, ENCFF580QGA, and from CISTROME 53597, 5388, 5389, 4940, 36310, 36316, 4971. For another GEO dataset, GSM2628088, reads were mapped to the hg19 build of the human genome with Bowtie2[63] and RELA peaks were identified with Macs2[64]. Alternative and Constitutive spliced exons were obtained from FasterDB[54]. To focus on intragenic RELA peaks, we used the bedtools[65] intersect command to remove all intergenic RELA peaks and all RELA peaks localized on first exon (or at least at less than 500 nt distance) for each gene. A Perl script was specifically created to measure the distance between RELA peaks and Tax-regulated exons. Briefly, RELA peaks and exons are provided as BED files, and the script reports for each exon the distance in nucleotides of the nearest RELA peak. Closest peak distances from the 710 Tax-regulated exon-cassettes were compared to closest peak distances from 710 exons chosen by chance ($10^5$ runs). Using bootstrap analysis, the random samples were compared with the set of Tax-regulated exons, and the normal distribution of these counts was used in a sample *t*-test to assess the significance of the RELA-binding sites enrichment at the vicinity of Tax-regulated exons. $p = 2e - 4$ was considered significant.

**TALE design and construct**. The TALE constructs were obtained from ThermoFisher Scientific. TALEs were constructed using the Golden Gate Assembly method[40]. The RVDs HD, NI, NG, and NN were chosen to specifically recognize the nucleotides C, A, T, and G, respectively. The TALE-targeting CD44 v10 sequence was 5′-TCCAACTCTAATGTCAATC-3′. This TALE construct was fused to a V5 sequence and a SV40 NLS at its 5′ end and cloned in the *Not*I-*Hin*dIII fragment of the pXJ41 backbone plasmid. DDX17-WT and DDX17-K142R cDNA were obtained by PCR from pcDNA3-HA-DDX17 and pcDNA3-HA-DDX17-K142R and were cloned in the *Hin*dIII–*Bgl*II fragment in the MCS downstream to the TALE sequence.

**CRISPR design and construct**. The sequence-specific sgRNA for site-specific interference of genomic targets were designed using CRISPRseek R package, and sequences were selected to minimize off-target effect[66]. Two complementary oligonucleotides were annealed and cloned into *Bbs*I site of pSpCas9(BB)-2A-Puro (PX459) V2.0 (Addgene plasmid #62988)[67] for co-expression with Cas9 using 5U of T4 DNA ligase, T4 DNA ligase buffer (1×) (Roche). 293T-LTR-GFP cells were transfected with the mix of equimolar ratio of PX459-sgRNA1 and PX459-sgRNA2 (Supplementary Data 5). At 24-h post-transfection, the medium was changed, 1 μg/ml puromycin was added for selection, and cells were cloned by serial dilution method.

**Reporting summary**. Further information on research design is available in the Nature Research Reporting Summary linked to this article.

## Data availability

RNA-seq data produced in this study have been deposited on NCBI GEO under the accession number GSE123752. The source data underlying Figs. 1c, 2b–f, 3a, b, d, 4a, c–f, 5a–d, 6a–f, and Supplementary Figs. 1, 2b–d, 3a, b, e, 4a–g, 5a–d, and 6a–c are provided as a Source Data file. All data supporting the findings of this study are available within the article and its supplementary information files and from the corresponding author upon reasonable request. Source data are provided with this paper.

## Code availability

FaRLine is publicly available (http://fasterdb.ens-lyon.fr/FARLINE.tgz). Other custom codes used in this study are available from the corresponding author upon reasonable request. Source data are provided with this paper.

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

## Acknowledgements

This study makes use of data (EGAS00001001296) generated by Department of Pathology and Tumor Biology, Kyoto University[24]. We gratefully acknowledge support from Laurent Modolo at the LBMC for bio-informatic advice, and from the PSMN (Pôle Scientifique de Modélisation Numérique) of the ENS de Lyon for computing resources. This work was supported by the Ligue Contre le Cancer (Comité de la Savoie, de la Drome et du Rhône), the Fondation ARC (ARC PJA20151203399), and the Agence Nationale pour la Recherche (program EPIVIR and CHROTOPAS). L.B.A. was supported by Ligue Contre le Cancer; G.G., by ANR CHROTOPAS; M.T., by a bursary from the French Ministry of Higher Education and Science; F.M., C.F.B, and D.A., by INSERM; and E.W., by Hospices Civils de Lyon and Lyon I University (France).

## Author contributions

Conceptualization, L.B.A., D.A., and F.M.; resources, A.G. and E.W.; experiments, L.B.A., P.M., M.T., and G.G.; technical support, E.C. and M.B.; bioinformatics, J.B.-C., H.P., N.F., and S.L.; formal analysis, L.B.A., M.T., G.G., and F.M.; supervision, D.A. and F.M.; funding acquisition, C.F.B., D.A. and F.M.; writing—original draft, F.M.; writing—review and editing, L.B.A, M.T., G.G., C.F.B., D.A., and F.M.

## Competing interests

The authors declare no competing interests.
