## [Peer Review File · Nature Communications]

Reviewers' Comments:

Reviewer #1:

Remarks to the Author:

Ben Ameer et al

The manuscript reports that HTLV-1 Tax recruits RelA to intragenic sites, which recruit splicing regulator DDX17 leading to alternative splicing. The idea that an oncoprotein may regulate alternative splicing is interesting and novel. However, there are numerous problems with the manuscript:

- 1) Most of the experiments were performed in 293T cells that are not physiologically relevant. It is unclear whether the same results would be obtained in CD4+ cells and in HTLV-1 infected cell lines and primary CD4+ cells.
- 2) Most of the experiments use Tax over-expression, resulting in levels likely to be much higher than those in HTLV-1 infected cells.
- 3) Many of the experiments show effects on splicing that are barely significant with 1.5x changes or less in the inclusion of an alternative exon.
- 4) It is stated that 48% of Tax-induced splicing modifications were detected at least once in 55 clinical samples. The clinical samples include asymptomatic carriers and ATLL subjects. However, many ATLL samples do not exhibit Tax expression. It is unclear which clinical samples were used in this analysis. The occurrence of such splicing abnormalities in 1 case is not very convincing, and the occurrence of only 48% of clinical samples is also unimpressive.
- 5) The use of qualitative rather than quantitative RT-PCR for examining levels of alternative spliced transcripts is problematic.
- 6) The results in Fig 4 are very confusing and appear to contradict the hypothesis. Deletion of the intragenic RelA binding site appears to have no effect on splicing (Fig 4C).
- 7) The manuscript includes a huge amount of data that are very poorly described.
- 8) The English grammar is very, very poor throughout the manuscript.

Reviewer #2:

Remarks to the Author:

This is a very interesting manuscript looking at the crosstalk between transcriptional regulation and splicing. The authors show how a transcriptional regulator from the NFkB signaling pathway, called RELA, regulates splicing by recruiting DDX17 nearby alternatively spliced exons dependent on this RNA helicase. Even more, the authors demonstrate the role of RELA in regulating DDX17-dependent splicing by targeting RELA to CD44v10 exon using TALENs. Finally, by recruiting TALE-DDX17 or an helicase-dead mutant, the authors demonstrate that this mechanism is dependent on the helicase activity and that DDX17 is the downstream effector and RELA the way DDX17 can be recruited nearby dependent exons.

The manuscript is clear. The data is solid and it is amongst the first ones to show a direct and causative role for a transcriptional regulator in regulating splicing via recruitment of an RNA regulator.

I will just recommend a couple of experiments (minor revision) before acceptance.

1) According to the model, RELA, via TAX expression, induces recruitment of DDX17, which in turn favors exon inclusion in 59% of TAX-dependent events. However, in Fig 1C, not all the exons are more included upon TAX expression and in Fig.3D, only 2/7 (29%) are affected upon knock down of DDX. It is true that 41% of TAX-dependent splicing events are not DDX dependent. Yet it would be nice to verify genome-wide whether it is true that TAX, RELA and DDX17 regulate splicing in the same direction and how many splicing events go with this rule. How many TAX-dependent exons are RELA dependent? How many TAX and RELA-dependent exons are DDX dependent? Do they go in the same direction? If not which proportion?

2) Some more genome-wide analyses would be nice too. For instance, is there a RELA motif amongst DDX-dependent exons? What is the Gene Ontology of DDX+TAX dependent spliced genes? Is it the same as TAX-dependent genes (see Fig1E)?

3) In the Hierarchical clustering of TAX-dependent exons (Fig.1D), there are only 3 uninfected CD4+ T-cells and AC are not really discriminated from ATLL. I think it is an overstatement to say that AC and ATLL samples can be discriminated from uninfected CD4+ T-cells based on this data.

4) Even though it is pretty clear that neither TAX nor RELA have a transcriptional effect on DDX-dependent spliced genes, we should be able to verify this in supplementary data every time TAX, RELA or DDX levels are affected to test splicing.

5) In FigS3B, the authors show that available RELA ChIP-seq datasets reveal that intragenic RELA peaks are significantly closer to alternative exons than to constitutive exons. What about comparing RELA levels in TAX-dependent exons vs alternatively spliced but not TAX-dependent exons?

6) I don't understand Fig4D. Where is the red line (aka TAX-regulated exons)? Are all the exons exactly at 1079bp?

7) In Fig.4H-I it is shown that without DDX, RELA cannot bind to DNA upon TAX expression. What about the other way around? In the absence of RELA, can DDX be recruited to the RNA? Another "control" would be to test the effect on splicing in the same conditions as tested in Fig.4I.

8) Then DDX17 is an RNA helicase. Can it bind to DNA? All the experiments performed are ChIPs. What about if the same was done but with RIP? I know that DDX17/RELA interaction is RNA independent. Yet, it will be nice to verify that the binding to the RNA is also RELA dependent, particularly when the binding to the chromatin is not exactly at the exon.

9) Fig 5 is very interesting, upon induced recruitment of TALE-RELA, DDX17 is also recruited. However upon TALE-DDX, RELA is not recruited, but splicing is affected. Seems like the role of RELA is to recruit DDX17. If this is true, the M22 splicing phenotype observed in Fig3D should be rescued with the TALE-DDX construct. This is a nice proof of the model.

10) The effect on splicing is very small (1,5xfold or less, Fig.5A). Is this biologically meaningful? Cell adhesion properties could be tested as in Fig.S1D.

SEP

SEP

Reviewer #3:

Remarks to the Author:

In this study the authors show that activation of the transcription factor NF- κ B by the viral oncogene TAX affects not only mRNA transcription of several hundred genes but also induces massive changes in alternative splicing. Their observations indicate a direct involvement of RELA in alternative splicing regulation mediated by co-recruitment of RELA and the splicing regulator DDX17 to intragenic regions. These findings extend the current view of NF- κ B-related gene expression control. Most of the presented data are of good quality and need only minor improvements. Nevertheless, important questions arise, which have to be addressed before the manuscript is suitable for publication in Nature Communications.

Comments in detail:

Major concerns:

1. The authors describe that TAX induced NF- κ B activation can cause changes in alternative splicing. In Figure 2 they demonstrate a physical interaction between TAX, DDX17 and RelA. Experiments with a TAX mutant form, defective in IKK activation, indicate that the observed interaction can be also driven by other stimuli. Moreover, recruitment of RELA to intragenic regions has been observed in other studies using different stimuli. These findings raise the question, if RelA-DDX17- driven alternative splicing actually requires TAX at all, as already indicate by Figure 5A. It should be tested in more detail, if the interaction between DDX17 and RELA can occur independently of TAX in situations where IKK is activated. Moreover, authors should treat cells with classical NF- κ B stimuli, such as TNF α or IL-1 β and subsequently analyze the occurrence of alternative splicing events in comparison to TAX induced splicing.
2. Although changes in alternative splicing of CD44 has not been identified in the initial screen (Figure 1 A and Table S1), authors emphasize subsequently on alternative splicing of CD44 (Figures 1C, S1 and 4). This seems to be inconsistent and needs further explanation.
3. The fact that TAX induces a large number of splicing events raises the question, how these changes affect physiological functions of the activated cells. At the current stage of the manuscript this issue is not addressed appropriately. In many cases, Tax-induced alterations appear quantitatively very moderate (e.g. Figure 1C). Further investigations showing phenotypic or functional alterations due to alternative splicing events are urgently needed.
4. In Figure 4, the authors describe that knockdown of siDDX5/17 affects recruitment of RELA to intragenic regions. Using the same experimental setting authors should also analyze recruitment of RELA to promoter regions to ensure that DDX5/17 affect recruitment specifically near splice sites.
5. In Figure 5 analyze chromatin and splicing regulation upon Tale-mediated tethering of RELA and DDX17. Results from Figure 5A raise the question, why p65 is recruited to DNA in unstimulated cells. The experimental setting indicates an experimental artefact due to ectopic overexpression of the protein. Experiment should also be done with stimulated cells. Moreover, in all experiments controls are missing, showing expression levels of the transfected TALE-protein constructs. Finally, the design and labeling for figures 5A-C is misleading.

Minor concerns:

1. Molecular weight markers are missing in Figures 2B-G and 3A-B
2. Labeling in Figure 4D needs to be checked. The size unit (kb) for "distance to RelA peak" seems to be wrong.
3. To better understand the meaning of Figure S2A, a more detailed description is required.
4. Figure S2B is not mentioned within the manuscript.
5. In line 176 of the manuscript authors mention Figure S2D. This figure does not appear in the current manuscript.

Taken together, in the present form this study is not suitable for publishing in Nature Communications.

Reviewer #4:

Remarks to the Author:

This is an interesting and provocative manuscript that identifies a new role for the RELA NF- κ B subunit, when activated by the HTLV-1 protein TAX, as a regulator of alternative mRNA splicing. The authors identify RELA dependent recruitment of the RNA helicase DDX17 to intragenic regions as the key regulatory step.

Overall the experiments are well performed and convincing. However, there are some areas where the manuscript would benefit from additional experiments to strengthen the overall conclusions of the authors and demonstrate a broader significance of the findings.

General comments

(1) All the mechanistic experiments are performed in a HEK 293 cells line (293T-LTR-GFP), where TAX was transiently over-expressed. Although these results were related back to splicing events in HTLV-1 infected cells it is not clear how general an effect this is. Can the authors demonstrate using ChIP the recruitment of RelA and DDX17 to the same intragenic regions in other cells types and ideally, if feasible, HTLV-1 infected cells?

(2) It is unclear what mechanistic role Tax is playing in this mechanism. Is it only required to activate NF- κ B and get RELA to the nucleus? In which case, would similar recruitment of RELA and DDX17 to intragenic regions and regulation of splicing be seen following TNF stimulation? However the co-IP data in Figure 2, implies a physical interaction between Tax and the RELA/DDX17 complex. Is Tax also recruited to the intragenic regions binding the RELA/DDX17 complex?

Other comments

(3) Is DDX17 recruitment to intragenic regions affected by siRNA depletion of RELA?

(4) In Fig 4C, the effect of the delta kB deletion on CD44 expression itself should be shown.

(5) The figure legends should have information on the cell type used for the analysis.

Point-by-point response to the referees' comments (NCOMMS-19-09246A)

We would like to thank the reviewers for their insightful comments and criticisms, which we believe have really helped us to improve the quality of the manuscript. We now provide additional experiments in the revised manuscript, including experiments using HTLV-1 infected cells, a deeper insight into molecular mechanisms involving TAX/RELA/DDX17 in regulating alternative splicing, and additional controls. Overall, the revised manuscript includes 9 new panels in the main Figures and 13 new panels in Supplementary Figures. With this, we believe that we have adequately addressed all the questions and comments raised by the Reviewers.

Reviewer #1 (Remarks to the Author):

The manuscript reports that HTLV-1 Tax recruits RelA to intragenic sites, which recruit splicing regulator DDX17 leading to alternative splicing. The idea that an oncoprotein may regulate alternative splicing is interesting and novel. However, there are numerous problems with the manuscript:

We thank the reviewer for his/her time and expertise in reviewing the manuscript and are pleased that he/she considers the role of the oncogene TAX in splicing regulation to be interesting.

1) Most of the experiments were performed in 293T cells that are not physiologically relevant. It is unclear whether the same results would be obtained in CD4+ cells and in HTLV-1 infected cell lines and primary CD4+ cells.

In order to address the reviewer's concern, we performed additional experiments in which we compared HTLV-1 infected CD4+ cell lines (C91PL and ATL2) to non-infected MOLT4 cells (Fig. 4i-k, **page 10, line 2** of the revised manuscript). We showed that the splicing patterns of a subset of genes in HTLV-1 infected cells are similar to the TAX-induced splicing pattern in HEK cells (Fig. 4j). In addition, both RELA and DDX17 were recruited in the vicinity of the regulated exons at the DNA level in the HTLV-1 infected cell (Fig. 4k). It is interesting to note that ATL2 cells expressed a higher level of TAX mRNA as compared to C91PL cells (Fig. 4i); accordingly, splicing variations were higher in ATL2 cells as compared to C91PL cells (Fig. 4j). We were not able to address this relationship in primary cells as experimental approaches (qChIP) required a prohibitively large number of cells. Nevertheless, these additional data are in agreement with the results presented in the first version of the manuscript showing that TAX-related splicing modifications are detected in naturally-infected CD4+ cells harvested from patients with HAM/TSP or ATLL (Fig. 1d and Supplementary Fig. 3a-b).

2) Most of the experiments use Tax over-expression, resulting in levels likely to be much higher than those in HTLV-1 infected cells.

To address this issue, we measured TAX mRNA levels in TAX-transfected HEK293T cells and in two HTLV-1 infected cell lines (ATL2 and C91PL). While the TAX mRNA level is similar in TAX-transfected HEK293T cells and C91PL cells, we observed that TAX mRNA levels are much lower in TAX-transfected HEK293T than in ATL2 cells (Fig. 4h). It is also important to underscore that TAX mRNA levels widely fluctuated in naturally infected CD4+ cells harvested from HAM/TSP patients (as illustrated in Supplementary Fig. S3B). This is consistent with previous reports indicating wide ranges of TAX expression in infected CD4+ clones derived from HAM/TSP patients (from 250×10^{-5} to 603,475 AU between infected cellular clones; median, 35,208; mean +/- SEM, 128,480 +/- 29,123) (Sibon, *et al.* J Clin Invest, 2006). More importantly, our results support a model where the role of TAX in alternative splicing regulation is mostly due to the activation of the NF- κ B pathway and the RELA-DDX17 axis (see Fig. 5 and below). This is important to keep in mind in order to not over-estimate the role of TAX in splicing regulation.

3) Many of the experiments show effects on splicing that are barely significant with 1.5x changes or less in the inclusion of an alternative exon.

We respectfully disagree with the reviewer's comment, as we have reported several splicing variations that can reach up to 40-fold (see for example Fig. 3d). Some small amplitude of splicing variations can be attributed to technical issues linked to the transient transfection assays, such as the proportion of cells receiving the expression vectors. For TAX-expression in parental and CD44 Δ kB cells, the selection of cells modified by CRISPR/Cas9 involved a high number of cell passages, which is known to reduce the cell transfection efficiency. Similarly, for TALE assays, this approach used large plasmids that are difficult to deliver in cells. We further

noticed that tethering either TALE-RELA or TALE-DDX17 to the genomic v10 exon led to affect cell adhesion of HEK cells at 48 h but not at 24 h post-transfections. As a result, qChIP and exon-specific RT-qPCR assays were carried out at 24 h for TALE assays (as compared to 48 h post-transfection for TAX). This has now been stated in the Method section (**page 23, line 16**). Using such short time delay experiments might explain lower variations in splicing. Nevertheless, we want to point out that splicing variations are still expressed, as noted by the rate of exon inclusion relative to the overall transcript level. Notably, we presented data as average values of at least 3 independent experiments, and we used statistical tests to assure the statistic robustness of biological variations. As an alternative approach, we also used RT-PCR assays for validating the percentage of exon inclusion rate predicted by RNA-seq analysis. Although less quantitative, this method is commonly used in splicing analyses, and it clearly ascertained the splicing modifications that had been observed by exon-specific qRT-PCR assays (Fig. 1c, Supplementary Figs. 2b and 5b).

It is also important to underline that splicing is a complex process involving a large number of complexes and proteins. For example, while it is well established that recruitment of DNA-binding proteins or changes in chromatin organization can affect splicing, the observed effects are generally small as other downstream rate-limiting steps are likely also playing a role (such as spliceosome recruitment and assembly, or RNA-binding protein recruitment). Thus, if the recruitment of DDX17 at the exon level on DNA has an effect on splicing (Fig. 5b), the observed effect may appear small in terms of magnitude because another downstream step in the splicing process is limiting. What is important is that if a DDX17 mutant is recruited using the same experimental approach, then no effect on splicing is observed (Fig. 5c). Furthermore, while the local recruitment of DDX17 induces CD44v10 inclusion (Fig. 5b), its depletion induces its exclusion upon TAX expression (Fig. 3d).

4) It is stated that 48% of Tax-induced splicing modifications were detected at least once in 55 clinical samples. The clinical samples include asymptomatic carriers and ATLL subjects. However, many ATLL samples do not exhibit Tax expression. It is unclear which clinical samples were used in this analysis. The occurrence of such splicing abnormalities in 1 case is not very convincing, and the occurrence of only 48% of clinical samples is also unimpressive.

We thank the reviewer for his/her comments, and we apologize for the lack of clarity in this part. In Table S1, we included the description of the previously published primary samples in dataset EGAS00001001296 (Kataoka, *et al.* Nat Genet, 2015). To address the reviewer's comment regarding the expression of TAX in ATLL, we now show that TAX is detected in most of the samples used (Supplementary Fig. 1). It is important to underscore that recently reports show that, although rarely detected in ATLL, TAX can be expressed in bursts, which may explain the TAX-like features of ATLL samples recently described *in vivo* (Kataoka, *et al.* Nat Genet, 2015; Billman, *et al.* Wellcome Open Res, 2017; Mahgoub, *et al.* Proc Natl Acad Sci U S A, 2018). We also would like to again emphasize (see Point 2) that, accordingly to our model, TAX induces splicing variations by activating the NF- κ B pathway. In this setting, ATL cells, irrespective of their TAX expression levels, globally exhibit high-level constitutive NF- κ B activation (Yamagishi, *et al.* Cancer Cell, 2012).

We have also to mention that we actually anticipated that ATLL samples would not express all the TAX-induced ASEs given the well-described high fluctuations of gene expression between individuals, proviral load, infected cells clonality, as well as somatic alterations (gene mutations, deletions, duplications...) that accumulate in ATLL cells. Conversely, we were surprised to recurrently identify some TAX-related ASEs in a large majority of infected samples as compared to donor CD4+ cells (e.g., the heatmap in Fig. 1d, Table S1). We believe these data are interesting as this is the first discovery of recurrent splicing alterations *in vivo* in a large set of ATLL samples. We have now included a new sheet in Table S1 that provides the inclusion rate values of exons regulated in the ATLL samples, to help colleagues to potentially focus on recurrent splicing events.

5) The use of qualitative rather than quantitative RT-PCR for examining levels of alternative spliced transcripts is problematic.

With the exception of Fig. 1c, all effects on splicing were measured by RT-qPCR in the revised version of the manuscript (Figs 3d, 4a, 4d, 4j, 5a-f). Nonetheless, in the splicing community, exon-specific RT-PCR analysis remains a typical and widely used/published method for validating RNA-seq predictions of ASE.

6) The results in Fig 4 are very confusing and appear to contradict the hypothesis. Deletion of the intragenic RelA binding site appears to have no effect on splicing (Fig 4C).

We apologize for the lack of clarity in that part. The term “intragenic” was used throughout the manuscript to qualify regions away from the promoter, i.e. localized downstream of the promoter in gene bodies. Concerning the gene CD44 (as illustrated in Fig. 4b), we carried out a CRISPR/Cas9-mediated deletion of the RELA binding site localized in the promoter region, not in the gene body. This approach allowed us to demonstrate that disturbing RELA recruitment to the promoter had no consequences either on the intragenic occupancy of RELA near the genomic exon v10 or on the inclusion rate of v10 at the RNA level. This result, together with others presented in the revised manuscript, confirms the role of intragenic enrichment of RELA in splicing regulation of CD44.

7) The manuscript includes a huge amount of data that are very poorly described.

We improved the description of data throughout the revised manuscript.

8) The English grammar is very, very poor throughout the manuscript.

The revised manuscript has been edited by native English speaker.

Reviewer #2 (Remarks to the Author):

This is a very interesting manuscript looking at the crosstalk between transcriptional regulation and splicing. The authors show how a transcriptional regulator from the NFκB signaling pathway, called RELA, regulates splicing by recruiting DDX17 nearby alternatively spliced exons dependent on this RNA helicase. Even more, the authors demonstrate the role of RELA in regulating DDX17-dependent splicing by targeting RELA to CD44v10 exon using TALENs. Finally, by recruiting TALE-DDX17 or an helicase-dead mutant, the authors demonstrate that this mechanism is dependent on the helicase activity and that DDX17 is the downstream effector and RELA the way DDX17 can be recruited nearby dependent exons. The manuscript is clear. The data is solid and it is amongst the first ones to show a direct and causative role for a transcriptional regulator in regulating splicing via recruitment of an RNA regulator.

I will just recommend a couple of experiments (minor revision) before acceptance.

We thank the reviewer for his/her positive comments and for valuable suggestions. His/her constructive comments, although stated as minors, significantly helped us to improve the manuscript.

1) According to the model, RELA, via TAX expression, induces recruitment of DDX17, which in turn favors exon inclusion in 59% of TAX-dependent events. However, in Fig 1C, not all the exons are more included upon TAX expression and in Fig.3D, only 2/7 (29%) are affected upon knock down of DDX. It is true that 41% of TAX-dependent splicing events are not DDX dependent. Yet it would be nice to verify genome-wide whether it is true that TAX, RELA and DDX17 regulate splicing in the same direction and how many splicing events go with this rule.? How many TAX and RELA-dependent exons are DDX dependent? Do they go in the same direction? If not which proportion?

We thank the reviewer for his/her comment requesting more detailed data. As shown in Table S1, TAX indeed promotes both positive and negative effects on various type of alternative splicing events, including skipping events, multi-exon skipping events, alternative donor and acceptor events, and mutually exclusive events. As requested by the reviewer, the revised Table S3 provides more details on the number, proportion, and direction of splicing events affected by TAX and siRNA DDX5/17. The data show that splicing modifications of TAX-regulated exons upon DDX5/17 depletion occur in an opposite way for 60%, 100%, 100%, and 60% of TAX-regulated alternatively skipped exon, donor, acceptor, and multi-exon skipping events, respectively. These data indicate that both positive and negative effects of TAX on splicing events can be affected by silencing DDX5/17 expression. This is in accordance with the dual functions of RNA helicases in splicing, which rely in part on their ability to unmask RNA motifs recognized by either positive or negative splicing regulators.

We understand the reviewer's interest for more RNA-seq analyses of samples invalidated or not for DDX5/17, TAX, or RELA expressions; however, these questions would require experimental approaches that are, in our opinion, beyond the scope of the present study, which was oriented to assess the molecular mechanisms underlying TAX- and RELA-associated splicing modifications.

2) Some more genome-wide analyses would be nice too. For instance, is there a RELA motif amongst DDX-dependent exons? What is the Gene Ontology of DDX+TAX dependent spliced genes? Is it the same as TAX-dependent genes (see Fig1E)?

We thank the reviewer for his/her constructive suggestion. Because CHIP-seq analyses have revealed a large number of RELA binding sites that do not correspond to RELA motifs (Martone, *et al.* Proc Natl Acad Sci U S A, 2003; Lim, *et al.* Mol Cell, 2007), we consider CHIP-seq data more informative than *in silico* predictions of NF-κB motifs. In this setting, we found that FasterDB-annotated alternative exons regulated by TAX and DDX5/17

corresponded to those intragenic regions more enriched in RELA-binding sites than their counterparts not regulated by TAX. These data have now been included in the revised manuscript (**page 9, line 14** and **Fig. 4d**).

To address the reviewer's comment regarding the gene ontology of DDX+TAX dependent spliced genes, we have now included a GO analysis of DDX+TAX dependent spliced genes (Supplementary Fig. 3c). The term Focal Adhesion was shared between TAX-dependent and TAX/DDX17/5-dependent spliced genes. Therefore, the RNA helicases may play an important role in mediating TAX biological effects on cell adhesion, as exemplified by the CD44 gene.

3) In the Hierarchical clustering of TAX-dependent exons (Fig.1D), there are only 3 uninfected CD4+ T-cells and AC are not really discriminated from ATLL. I think it is an overstatement to say that AC and ATLL samples can be discriminated from uninfected CD4+ T-cells based on this data.

We apologize for the lack of clarity in this part. Accordingly, we have clarified this point in the revised manuscript (**page 5, line 21**). As mentioned to the reviewer #1 (answer #4), we actually anticipated that ATLL samples would not express all the TAX-induced ASEs given the well-described high fluctuations of gene expression between individuals, proviral load, infected cell clonality, as well as somatic alterations (gene mutations, deletions, duplications...) that accumulate in ATLL cells. Conversely, we were surprised to repeatedly identify some TAX-related alternative splicing events in a large majority of infected samples as compared to donor CD4+ cells (see heatmap in Fig. 1d, Table S1). We believe that these data are of high interest, as this is the first discovery of recurrent splicing alterations *in vivo* in a large set of ATLL samples. We included now a new sheet in Table S1 in the revised manuscript with the inclusion rate values of exons regulated in the ATLL samples, to help colleagues potentially focus on recurrent splicing events.

4) Even though it is pretty clear that neither TAX nor RELA have a transcriptional effect on DDX-dependent spliced genes, we should be able to verify this in supplementary data every time TAX, RELA or DDX levels are affected to test splicing.

As pointed out by the reviewer, our results suggest that the splicing effects of TAX, DDX17, and RELA occur independently of transcriptional effects. This was first suggested by our RNA-seq analyses (Fig. 1a), which showed that only a minority of genes (3.5%, 33/939) are altered at both expression and splicing levels upon TAX expression. We then confirmed these data by additional exon-specific RT-qPCR analyses of 6 selected exons (Fig. 3d) that revealed that TAX induced significant TAX-induced exon inclusion events but only had weak and variable effects on global levels of corresponding RNA. According to the reviewer's request, we have now provided additional controls showing that the siRNAs siDDX17/5 and siRELA did not markedly influence the expression level of those genes (Supplementary Fig. 2d). In contrast, and in line with previous work suggesting that NF- κ B directly regulates the CD44 promoter (Smith, *et al.* PLoS One, 2014), we noticed a slight reduction of CD44 expression in TAX-expressing cells knocked down for RELA expression (Supplementary Fig. 4b). We thus considered such regulations as an opportunity to address whether or not RELA-dependent transcription and splicing are two interrelated processes. Consequently, we demonstrated with CRISPR/Cas9-mediated approaches that the RELA chromatin occupancy of the CD44 promoter affected the global gene expression of CD44 but not the intragenic recruitment of RELA or the corresponding splicing regulation of exon v10 (Fig. 4b-c). This explanation has now been included (**page 8, line 23**) and the gene expression controls have been mentioned throughout the revised manuscript.

5) In FigS3B, the authors show that available RELA ChIP-seq datasets reveal that intragenic RELA peaks are significantly closer to alternative exons than to constitutive exons. What about comparing RELA levels in TAX-dependent exons vs alternatively spliced but not TAX-dependent exons?

To address this question, we carried out additional genome-wide analysis of the chromatin distribution of RELA. The results showed that TAX-regulated exons displayed higher RELA enrichment at their vicinity than alternative exons not affected by TAX (Fig. 4e). This observation has now been included (**page 9, line 14**) in the revised manuscript.

6) I don't understand Fig4D. Where is the red line (aka TAX-regulated exons)? Are all the exons exactly at 1079bp? More explanations.

We apologize for the lack of clarity in this figure. The red line corresponds to the median distance between TAX-regulated exons and the closest RELA binding site. More explanations have been included in the Method section of the revised manuscript (**page 25 line 11**). To summarize, closest peak distances from the 710 TAX-regulated exon-cassettes were compared to closest peak distances from 710 exons chosen by chance in an iterative manner (10^5 runs). The random samples were compared with the set of TAX-regulated exons using a bootstrap analysis and the normal distribution of these counts was used in a sample t-test to assess the significance of the RELA-binding sites enrichment at the vicinity of TAX-regulated exons.

7) In Fig.4H-I it is shown that without DDX, RELA cannot bind to DNA upon TAX expression. What about the other way around? In the absence of RELA, can DDX be recruited to the RNA? Another "control" would be to test the effect on splicing in the same conditions as tested in Fig.4I. (RT-PCR ? as Figure 3)

We thank the reviewer for these constructive proposals. We attempted to carry out RIP assays to assess whether or not the depletion of RELA could affect the RNA-binding of DDX17. However, due to a very high background of RNA-bound DDX17, this analysis was not sufficiently reliable (not shown). This was probably due to free DDX17 proteins that are capable of non-specific binding to RNA in cell lysates.

To ascertain the involvement of RELA in DDX17 recruitment and subsequent splicing regulation, we carried out additional qChIP analyses of RELA and DDX17 in HEK cells that did or did not express TAX and were knocked down or not for RELA expression by siRNA. The results revealed that decreased gene expression of RELA affected DDX17 chromatin binding at TAX splicing targets, and in turn abolished their alternative splicing regulation (Figs. 3d and 4h). This is consistent with our previous results obtained in TALE assays that showed that RELA acts as a chromatin anchor of DDX17 (Fig. 5). These new results have now been included in the revised manuscript (**page 7, line 25 and page 9, line 33**).

8) Then DDX17 is an RNA helicase. Can it bind to DNA? All the experiments performed are ChIPs. What about if the same was done but with RIP? I know that DDX17/RELA interaction is RNA independent. Yet, it will be nice to verify that the binding to the RNA is also RELA dependent, particularly when the binding to the chromatin is not exactly at the exon.

Both DDX17 and DDX5 are composed of an RG-rich region, adjacent to the RNA helicase domain, that can function in RNA binding, protein-protein interactions, and/or protein localization. They do not contain a DNA binding domain but interact with and regulate various DNA-binding factors, including transcription factors. Owing to this property, the RNA helicases DDX5 and DDX17 have been previously identified as important co-factors in promoter regulation (Fuller-Pace *Biochim Biophys Acta*, 2013; Bourgeois, *et al. Nat Rev Mol Cell Biol*, 2016). As many RNA helicases can bind to double-stranded RNA, we can not exclude that they also bind DNA directly. However, in the biological system described in this manuscript, our results suggest that DDX17 is recruited by the RELA DNA binding activity, as depletion of RELA resulted in the decrease of DDX17 recruitment to DNA (Fig. 4h).

As already mentioned, our attempt to analyze the DDX17–RNA interaction did not provide reliable results. We would like to point out that setting up the TALE assay showing the role of DDX17 recruitment at the DNA level on

splicing outcome was already a very challenging task and is providing original results (Fig. 5a-c). Accordingly, in our opinion, the more important message of the present article is the new role of RELA in providing splicing target specificity by anchoring DDX17 to the chromatin in the vicinity of exon targets, which are dynamically defined upon NF- κ B signaling. TALE-assays revealed that the RNA helicase activity of DDX17 is required for regulating splicing of RELA-targeted exons; however, we believe that investigating the role of RELA in RNA binding of DDX17 is beyond the scope of the present study.

9) Fig 5 is very interesting, upon induced recruitment of TALE-RELA, DDX17 is also recruited. However, upon TALE-DDX, RELA is not recruited, but splicing is affected. Seems like the role of RELA is to recruit DDX17. If this is true, the M22 splicing phenotype observed in Fig3D should be rescued with the TALE-DDX construct. This is a nice proof of the model.

We thank the reviewer for suggesting this experiment that we carried out by transiently co-transfecting HEK cells using vectors encoding the M22 mutant and TALE-DDX17. Indeed, the control-like phenotype of M22-expressing cells was rescued to a TAX-like phenotype upon TALE-DDX17 expression (Fig. 5d, **page 11 line 18**). This confirmed that tethering DDX17 to RELA's target exon, irrespectively of full NF- κ B activation by TAX, is sufficient for regulating alternative splicing.

10) The effect on splicing is very small (1,5 xfold or less, Fig.5A). Is this biologically meaningful? Cell adhesion properties could be tested as in Fig.S1D.

We respectfully disagree with the reviewer's comment, as we are reporting several splicing variations that can reach up to 40-fold (see for example Fig. 3d). Some small amplitude of splicing variations can be attributed to technical issues linked to the transient transfection assays, such as the proportion of cells receiving the expression vectors. For TAX expression in parental and CD44 Δ kb cells, the selection of cells modified by CRISPR/Cas9 involved high number of cell passages, which is known to reduce the cell transfection efficiency. Similarly, TALE assays used large plasmids that are difficult to deliver in cells. We further noticed that tethering either TALE-RELA or TALE-DDX17 to the genomic v10 exon affected cell adhesion of HEK cells at 48 h but not at 24 h post-transfection. As a result, qChIP and exon-specific RT-qPCR assays were carried out earlier in TALE assays, at 24 h compared to 48 h post-transfection for TAX. Such a short time delay experiment may explain lower variations in splicing. This has now been commented on in the method section of the revised manuscript (**page 23**). Unfortunately, due to same technical limitations, we did not succeed in cell adhesion assays using TALE constructs, mainly because TALE-transfected cells did not survive trypsination and died before re-plating in hyaluronate- or collagen-coated substrates. However, we demonstrated that upon TAX expression HEK cells display a CD44v10 like-phenotype switching cell adhesion properties from Hyaluronate to Collagen coated substrates, thereby arguing for phenotypic consequences of TAX-induced splicing modifications (Supplementary Fig. 3e).

It is also important to underline that splicing is a complex process involving a large number of complexes and proteins. For example, while it is well established that recruitment of DNA-binding proteins or changes in chromatin organization can affect splicing, the observed effects are generally small because other downstream rate-limiting steps, such as spliceosome recruitment and assembly or RNA-binding protein recruitment, are likely also playing a role. Thus, if DDX17 recruitment to DNA at the exon level has an effect on splicing (Fig. 5b), the observed effect may appear small in terms of magnitude because another downstream step in the splicing process is limiting. What is important is our observation that if a DDX17 mutant is recruited using the same experimental approach, no effect on splicing is observed (Fig. 5c). Furthermore, while the local recruitment of DDX17 induced CD44 v10 inclusion (Fig. 5b), its depletion induced its exclusion upon TAX expression (Fig. 3d).

Nevertheless, we would like to point out again that splicing variations are expressed as the rate of exon inclusion relative to the overall transcript level. Our data are presented as average values of at least 3 independent experiments, and statistical tests are used to assess the statistic robustness of biological variations. As an

alternative approach, we also used RT-PCR assays for validating the percent exon inclusion rate predicted by RNA-seq analysis. Although less quantitative, this method is commonly used in splicing analyses; here, it clearly verified the splicing modifications indicated by exon-specific qRT-PCR assays (Fig. 1C, Supplementary Figs. 2b and 5b).

Reviewer #3 (Remarks to the Author):

In this study the authors show that activation of the transcription factor NF- κ B by the viral oncogene TAX affects not only mRNA transcription of several hundred genes but also induces massive changes in alternative splicing. Their observations indicate a direct involvement of RELA in alternative splicing regulation mediated by co-recruitment of RELA and the splicing regulator DDX17 to intragenic regions. These findings extend the current view of NF- κ B-related gene expression control. Most of the presented data are of good quality and need only minor improvements. Nevertheless, important questions arise, which have to be addressed before the manuscript is suitable for publication in Nature Communications.

We thank the reviewer for his/her positive and for his/her constructive in the improvement of the manuscript. We have revised our manuscript to fully address these comments and suggestions.

Comments in detail:

Major concerns:

1. The authors describe that TAX induced NF- κ B activation can cause changes in alternative splicing. In Figure 2 they demonstrate a physical interaction between TAX, DDX17 and RelA. Experiments with a TAX mutant form, defective in IKK activation, indicate that the observed interaction can be also driven by other stimuli. Moreover, recruitment of RELA to intragenic regions has been observed in other studies using different stimuli. These findings raise the question, if RelA-DDX17- driven alternative splicing actually requires TAX at all, as already indicate by Figure 5A. It should be tested in more detail, if the interaction between DDX17 and RELA can occur independently of TAX in situations where IKK is activated. Moreover, authors should treat cells with classical NF- κ B stimuli, such as TNF α or IL-1 β and subsequently analyze the occurrence of alternative splicing events in comparison to TAX induced splicing.

The reviewer is right: the main function of TAX in the splicing regulatory process reported here is to activate the NF- κ B pathway. We now show that TAX is dispensable for promoting RELA/DDX17-dependent splicing of v10 when cells are exposed to TNF α and PMA (phorbol 12-myristate 13-acetate) (**page 11, line 23**, Fig. 5e). More importantly, using ectopic expression of RELA, we could also reproduce this effect in a dose-dependent manner, thereby ascertaining the direct role of RELA in such splicing modification (Fig. 5e). These effects coincided with the dynamic formation of RELA:DDX17 complexes (Supplementary Fig. 5c). Altogether, these data are consistent with results obtained in TALE assays indicating that experimental tethering of RELA to genomic exons permits recruitment of DDX17 and regulation of Ev10 splicing in the absence of TAX.

2. Although changes in alternative splicing of CD44 has not been identified in the initial screen (Figure 1 A and Table S1), authors emphasize subsequently on alternative splicing of CD44 (Figures 1C, S1 and 4). This seems to be inconsistent and needs further explanation.

The fact that CD44 was not identified in our RNA-seq analysis is not surprising, as the CD44 splicing pattern is very complex and relies on 10 consecutive variable exons. The FARELINE pipeline that we developed to analyze RNA-seq datasets cannot handle this level of complexity, as FARELINE (like many other pipelines) relies on exon-exon junction reads. We anticipate therefore that the splicing modifications induced by TAX expression and HTLV-1 infection is more complex than what we are reporting. In this setting, we precisely analyzed the CD44 splicing pattern based on previous reports that reported its splicing variations in HTLV-1-infected cells (Thenoz, *et al.* *Retrovirology*, 2014; Matsuoka, *et al.* *J Neuroimmunol*, 2000). As we observed an effect of TAX on CD44 splicing, we decided to investigate the molecular mechanisms using this gene as a model because: i) many splicing variants affected by TAX come from genes involved in cell adhesion (Supplementary Fig. 3c); and ii) besides RELA-dependent splicing regulations, RELA depletion also affects CD44 expression in TAX-expressing

cells, which gave us the opportunity to address the interrelated relationship between RELA-related mechanism controlling transcription initiation and splicing. These explanations have been included in the revised manuscript (page 7, line 33 and page 8, line 24).

3. The fact that TAX induces a large number of splicing events raises the question, how these changes affect physiological functions of the activated cells. At the current stage of the manuscript this issue is not addressed appropriately. In many cases, Tax-induced alterations appear quantitatively very moderate (e.g. Figure 1C). Further investigations showing phenotypic or functional alterations due to alternative splicing events are urgently needed.

Indeed, TAX expression changes the splicing pattern of a large number of genes. In our point of view, this observation is strictly identical to the observation that TAX induces changes of the expression level of many genes. The question of which changes are really relevant with regard to functional consequences can arise for any study based on large-scale datasets. In this context, we believe that the results presented in this article provide interesting insights. Intriguingly, we observed that genes whose expression levels are modified by TAX are associated with terms such as "TNF signaling pathway", while genes regulated at the splicing level are associated with terms such as "focal adhesion" (Supplementary Fig. 3c). This suggests that TAX-regulated splicing events may play a particularly important role in changing the adhesion properties of infected cells that have been reported in many studies (Curis, *et al.* J Virol, 2016; Kim, *et al.* J Immunol, 2006; Sasaki, *et al.* Blood, 2005; Tanaka, *et al.* J Virol, 1996). Indeed, we show that adhesion of TAX-expressing cells is different from that of control cells (Supplementary Fig. 3e). In addition, using an exon ontology (EO) approach that we recently developed to estimate enrichment in protein features encoded by exons, we found that TAX- and DDX17-regulated exons affect regions involved in functionally validated post-translational modifications, protein structures, and binding functions (Supplementary Fig. 3C), which together are likely to affect the connectivity network between proteins and subsequently to contribute to modifying cell phenotype. This point has been included in the revised manuscript (page 8, line 3 and page 12, line 21).

Moreover, we wish to emphasize that the originality of our work resides in both: i) the demonstration that TAX expression induces splicing variations through RELA; and ii) the characterization of the mechanistic cascade involving RELA and DDX17. More specifically, we show for the first time that the RELA transcription factor that binds within gene bodies in the vicinity of exons can locally recruit proteins involved in splicing. The original approaches used in this work, such as the TALE assay, establish a model in which DNA-binding transcription factors not only regulate transcription by recruiting transcription coregulators at the chromatin level but also regulate splicing at the chromatin level by locally recruiting splicing-regulatory proteins. These mechanistic insights are important to keep in mind when evaluating the biological importance of our work.

4. In Figure 4, the authors describe that knockdown of siDDX5/17 affects recruitment of RELA to intragenic regions. Using the same experimental setting authors should also analyze recruitment of RELA to promoter regions to ensure that DDX5/17 affect recruitment specifically near splice sites.

We wish to emphasize that it is not unprecedented that the depletion of a coregulator (such as DDX17) can in turn destabilize an interaction between a transcription factor (here, RELA) and DNA. For example, our group has recently reported that DDX5/17 depletion impacts the interaction between the transcription factor REST and DNA, which identified DDX5/17 as REST transcriptional coregulators (Lambert, *et al.* Nucleic Acids Res, 2018). We now show that DDX17 depletion can affect the stability of the RELA interaction within genes as well as within the promoter (Supplementary Fig. 6a-c). As we still have no explanation for this phenomenon, and as this observation is not directly related to the model proposed in the paper, we have decided to present these data as supplementary data. Indeed, the main focus of the manuscript is to highlight that TAX activates RELA which recruits DDX17 on DNA with consequences on splicing. This point has been introduced in the discussion section (page 13, line 17).

5. In Figure 5 analyze chromatin and splicing regulation upon Tale-mediated tethering of RELA and DDX17. Results from Figure 5A raise the question, why p65 is recruited to DNA in unstimulated cells. The experimental setting indicates an experimental artefact due to ectopic overexpression of the protein. Experiment should also be done with stimulated cells in order to better address the direct effect of RELA on Exons. Moreover, in all experiments controls are missing, showing expression levels of the transfected TALE-protein constructs. Finally, the design and labeling for Figures 5A-C is misleading.

We apologize for the lack of clarity regarding the design of Fig. 5a. TALE-RELA was precisely designed to direct RELA binding to the genomic exon v10 of the CD44 gene in unstimulated cells (Fig. 5a). As TALE-RELA binds to DNA, there is no need for cell stimulation. Western blots showing the expression levels of the transfected TALE-protein constructs are now provided in Supplementary Fig. 5a. The labeling of Figures 5a-c has been modified.

Minor concerns:

1. Molecular weight markers are missing in Figures 2B-G and 3A-B

Molecular weight markers have been added to Figs 2, 3, and Supplementary Fig. 5.

2. Labeling in Figure 4D needs to be checked. The size unit (kb) for “distance to RelA peak” seems to be wrong. As requested, this has been corrected (Fig. 4f).

3. To better understand the meaning of Supplementary Fig. 2A, a more detailed description is required. More details have been added to the Figure legend.

4. Supplementary Fig. 2B is not mentioned within the manuscript.

We apologize for this error, which has now been corrected (**page 7, line 17**).

5. In line 176 of the manuscript authors mention Supplementary Fig. 2D. This Figure does not appear in the current manuscript.

We apologize for this error; it has been corrected.

Taken together, in the present form this study is not suitable for publishing in Nature Communications.

We hope that our significant revision has improved our manuscript enough to change the reviewer’s opinion.

Reviewer #4 (Remarks to the Author):

This is an interesting and provocative manuscript that identifies a new role for the RELA NF- κ B subunit, when activated by the HTLV-1 protein TAX, as a regulator of alternative mRNA splicing. The authors identify RELA dependent recruitment of the RNA helicase DDX17 to intragenic regions as the key regulatory step.

Overall the experiments are well performed and convincing. However, there are some areas where the manuscript would benefit from additional experiments to strengthen the overall conclusions of the authors and demonstrate a broader significance of the findings.

We thank the reviewer for his/her positive comments emphasizing the importance and quality of our work.

General comments

(1) All the mechanistic experiments are performed in a HEK 293 cells line (293T-LTR-GFP), where TAX was transiently over-expressed. Although these results were related back to splicing events in HTLV-1 infected cells it is not clear how general an effect this is. Can the authors demonstrate using ChIP the recruitment of RelA and DDX17 to the same intragenic regions in other cells types and ideally, if feasible, HTLV-1 infected cells?

As also requested by the reviewer 1, we have now extended our analyses to HTLV-1–infected CD4+ cell lines (namely, C91PL and ATL2) and confirmed the RELA/DDX17-dependent alternative splicing events in these cells as compared to the non-infected MOLT4 cells (Fig. 4h). In addition, we were able to show that both RELA and DDX17 bind to DNA in the vicinity of regulated exons in ATL2 cells when compared to the control MOLT4 cells (Fig. 4k). Interestingly, the magnitude of splicing regulations was higher in ATL2 than in C91PL cells, and it appeared to correlate with the higher expression level of TAX in ATL2 cells as compared to C91PL cells, which we believe resulted in a stronger activation of the NF- κ B signaling pathway (as suggested by the higher IL8 expression level in ATL2 cells as compared to C91PL cells) (Fig. 4h). Collectively, these results strengthen our conclusions (**page 9, line 33**).

(2) It is unclear what mechanistic role Tax is playing in this mechanism. Is it only required to activate NF- κ B and get RELA to the nucleus? In which case, would similar recruitment of RELA and DDX17 to intragenic regions and regulation of splicing be seen following TNF stimulation? However, the co-IP data in **Figure 2**, implies a physical interaction between Tax and the RELA/DDX17 complex. Is Tax also recruited to the intragenic regions binding the RELA/DDX17 complex?

We apologize for the lack of clarity regarding the mechanistic role of Tax in the reported mechanism (as also noted by reviewer 3). Accordingly, we now demonstrate that TAX is in fact dispensable for promoting RELA/DDX17-dependent splicing of v10 when cells are exposed to either TNF- α or PMA (Fig. 5e, **page 11, line 23**). More importantly, ectopic expression of RELA was the only factor necessary to reproduce this effect, thereby arguing for a direct role of RELA in this splicing modification. These data are consistent with the results obtained from TALE assays, which indicated that experimental tethering of RELA to genomic exons recruited DDX17 in the absence of TAX or RELA activation. Further, these data indicate that TAX binding to DNA is not required in this mechanism, although we cannot rule out such a possibility (as we did not obtain reliable results when performing TAX-qChIP, mainly due to non-specific interactions and high background signals in crosslinked cell lysates; data not shown). This point is now addressed in the discussion of the revised manuscript (**page 14, line 8**).

Other comments

(3) Is DDX17 recruitment to intragenic regions affected by siRNA depletion of RELA?

We now provide additional qChIP analyses of DDX17 in cells that do or do not express TAX and that have been invalidated or not for RELA expression (Fig. 4h and Supplementary Fig. 4g). The results indicate that RELA

silencing strongly affected the DDX17 occupancy of exons regulated by TAX (Fig. 4h), and in turn abolished splicing regulation at the RNA level (Fig. 3d), which further strengthen our conclusions. These new data have been included in the revised manuscript (**page 9, line 33**).

(4) In Fig 4C, the effect of the delta kB deletion on CD44 expression itself should be shown.

As requested, we provide now additional controls showing the effect of the Δ kb deletion on CD44 expression (Supplementary Fig. 4c, **page 9, line 6**).

(5) The Figure legends should have information on the cell type used for the analysis.

This has been corrected.

Reviewers' Comments:

Reviewer #2:

Remarks to the Author:

The authors addressed all my concerns.

Reviewer #3:

None

Reviewer #4:

Remarks to the Author:

I am happy with the revisions supplied by the authors and have no further comments.

Responses to the reviewers' comments:

Reviewer #2 (Remarks to the Author):

The authors addressed all my concerns.

We thank the reviewer for his/her constructive comments over the course of article revisions. We are glad to see that his/her comments have been addressed.

Reviewer #4 (Remarks to the Author):

I am happy with the revisions supplied by the authors and have no further comments.

We thank the reviewer for his/her effort in reviewing our manuscript. We are glad the reviewer was satisfied with our manuscript revision.